# Optimization of passive acoustic bedload monitoring in rivers by signal inversion

Mohamad Nasr[1], Adele Johannot[1], Thomas Geay[2,3], Sebastien Zanker[4], Jules Le Guern[.3], Alain Recking[1]

[1]University Grenoble Alpes, INRAE, ETNA, Grenoble, France.

[2] Office National des Forêts, service Restauration Terrain Montagne 38000 Grenoble, France.

[3] GINGER BURGEAP, R&D, 38000 Grenoble, France.

[4] EDF Hydro, DTG, 38950 Saint-Martin-le-Vinoux, France.

*Correspondence to*: Mohamad Nasr (mohamadnasr94@gmail.com)

**Abstract.**

Recent studies have shown that hydrophone sensors can monitor bedload flux in rivers by measuring the self-generated noise (SGN) emitted by bedload particles when they impact the riverbed. However, experimental and theoretical studies have shown that the measured SGN depends not only on bedload flux intensity but also on the propagation environment, which differs between rivers. Moreover, the SGN can propagate far from the acoustic source and be well measured at distant river positions without bedload transport. It has been shown that this dependency of the measured SGN data on the propagation environment

can significantly affect the performance of monitoring bedload flux by hydrophone techniques. In this article, we propose an inversion model to solve the problem of SGN propagation and integration effect. In this model, we assume that the riverbed acts as SGN source areas with intensity proportional to the local bedload flux. The inversion model locates the SGN sources and calculates their corresponding acoustic power by solving a system of linear algebraic equations accounting for the actual measured cross-sectional acoustic power (acoustic mapping) and attenuation properties. We tested the model using data from

measured bedload SGN profiles (acoustic mapping with a drift boat) and bedload flux profiles (direct sampling with an Elwha sampler) acquired during two field campaigns conducted in 2018 and 2021 on the Giffre River in the French Alps. Results confirm that the bedload flux measured at different verticals on the river cross-section correlates with the inversed acoustic power than measured acoustic power. Moreover, it was possible to fit data from the two field campaigns with a common curve after inversion, which was not possible with the measured acoustic data. The results of the inversion model, compared to

measured data, show the importance of considering the propagation effect when using the hydrophone technique and offer new perspectives for the calibration of bedload flux with SGN in rivers.

## 1 Introduction

Bedload transport controls rivers' morphodynamics and can directly impact population safety, hydraulic structures' stability, and river ecological systems. Meanwhile, bedload transport is a consequence of the morphology (Recking et al., 2016) as it

occurs at different rates across the channel (Gomez, 1991) due to heterogeneity in riverbed grains size distribution (GSD), riverbed geometry, flow depth, and velocity (Whiting and Dietrich, 1990; Ferguson et al., 2003). Understanding the transport

dynamics thus requires coupling of water flow gradient, river bed adjustment, and roughness conditions (Ergenzinger et al., 1994). This explains why estimating bedload transport and its impact on the riverbed is not an easy task. For instance, computation with bedload equations usually considers the average shear stress $\tau$, occulting the non-linear effect of variability within the section (Ferguson et al., 2003; Recking, 2013). On the other hand, direct monitoring of bedload transport (e.g., pressure difference samplers) is expensive and time-consuming and does not permit high spatio-temporal resolution sampling (Claude et al., 2012).

Given these difficulties, particular interest has been given to indirect surrogate bedload monitoring using different sensors (Gray et al., 2010). One category of these techniques is the passive sensing technique, which measures the signals emitted by bedload impacts. These techniques permit high-resolution monitoring even under extreme flow conditions. Bedload particles can impact an object specifically designed for this measurement; for instance, geophones are used to measure the vibration generated by particles' impacts on steel plates (Rickenmann et al., 2014), and microphones are used to measure the acoustic noise generated inside impacted steel pipes (Mao et al., 2016). Another approach directly measures the signal emitted when the transported grains hit the riverbed. For instance, seismometers measure ground vibrations due to bedload impacts (Gimbert et al., 2019a; Bakker et al., 2020), whereas hydrophones measure the bedload self-generated acoustic noise (SGN) (Johnson and Muir, 1969; Barton et al., 2010). This paper concerns this later technique.

Recent studies have shown that the measured SGN depends not only on bedload characteristics but also on the sound propagation properties of the river, which is controlled by multiple factors such as slope, water level, and bed roughness (Wren et al., 2015; Rigby et al., 2016; Geay et al., 2017). For example, in their attempt to derive a general calibration curve between bedload flux and acoustic power, (Geay et al., 2020) observed that the spectral content of SGN was highly correlated to the riverbed slope which is a parameter that significantly controls the propagation environment of the river (Geay et al., 2019). (Geay et al., 2020) then suggest the significant impact of the local propagation effect of the river on the measured SGN. This dependency of SGN on the local conditions may have contributed to the general scattering obtained between specific bedload flux and acoustic power in the mentioned work. On the other hand, this also suggests that accounting for propagation effects should improve the relationships between SGN and bedload characteristics. Besides, an inversion method that estimates the entire bedload GSD curve from the measured SGN spectrum has been proposed by (Petrut et al., 2018). However, the GSD inversion model tested on five gravel-bed rivers has overestimated the measured values in particular for the finest materials (Geay et al., 2018). The latest suggested that the acoustic power measured in rivers may not adequately capture the SGN the of finest materials contained in bedload due to signal attenuation at high frequencies.

The correction of signal attenuation due to propagation can be achieved by using source inversion methods. The inversion method uses propagation laws to reconstruct the strengths and location of sources from the measured signal. It is extensively studied and used in acoustical engineering applications such as detecting noise sources for jet engines using a beamforming microphone array by manipulating the phase and the amplitude of the wave form (Presezniak and Guillaume, 2010), identify acoustic emissions in machinery using the spectral analysis coupled with the time-domain of acoustic signals (Arthur et al., 2017), and analyze vibrational patterns in automotive components using finite element models to reconstruct the source and

propagation path (Madoliat et al., 2017). In seismology, inversion techniques have been instrumental in locating seismic sources using the amplitude source location (ASL) method (Battaglia and Aki, 2003; Walter et al., 2017), investigating microseismic events related to hydraulic fracturing using Stochastic inversion techniques (Maxwell, 2014), and understanding the structure of Earth's interior by determining the velocity distribution of the propagated waves (Rawlinson et al., 2010).

Regardless of the specific field, inversion methods inherently involve modeling the propagation of signals in different environments. However, the inversed parameters and the used algorithm can widely vary depending the studied domain and the specificity of each application .

In our work, the inversion is based on the spectral content of the measured bedload SGN signals propagated withing the river water column. To our knowledge, no studies have dealt with bedload SGN sources inversion in rivers. Despite, its evident

interest for bedload monitoring that inversion would give access to the characteristics of SGN sources which can improve our understanding of the bedload characteristics and distribution in the rivers. Recently, (Geay et al., 2019) proposed a protocol to estimate the acoustic signal attenuation in rivers using a transmission loss function (*TL*) calibrated with an active acoustic experiment.

In this paper, we use the work of Geay et al. (2019) function for developing an inversion model that gives access to the SGN

sources by correcting the attenuation of the measured SGN. First, we define the bedload SGN source and the transmission loss function in the river. Second, we present the inversion model adopted for SGN sources. Finally, we test the proposed model's performance in the field with two experiments: 1) an active test (in the river and the lab) using a known emitted signal, 2) a passive test using bedload SGN measurements where the inversed sources are compared with bedload physical sampling.

## 2 Theoretical definitions

**2.1 Bedload SGN source**

Acoustic noise corresponds to minute impulsive pressure p fluctuations initiated at the source position and propagated to different positions. In underwater acoustics, the pressure is typically measured in micro-pascals (µPa), which is the standard metric unit for this field, and will be the unit of choice used within this work. By convention, the intensity of an acoustic source is defined as the intensity measured at a distance of 1 m from the source without being attenuated (Jensen et al., 2011). Multiple

studies have examined the generation of acoustic noise by impacting body in the air (Koss and Alfredson, 1973; Koss et al., 1974; Akay et al., 1978). However, less research was dedicated to studying the acoustic noise generated by underwater sediment impacts (Thorne and Foden, 1988; Thorne, 1990). The physical model proposed by Thorne and Foden (1988) suggests a frequency-based solution of sound generated due to a sphere-sphere impact underwater. The model computes the energy spectrum which is the variation of acoustic energy (in square unit pressure $µPa^2$) per unit frequency (Hz) over a finite

period of time (in seconds). The model shows that the energy spectrum $e$ ($µPa^2 \cdot s \cdot Hz^{-1}$) of acoustic noise generated due to

acceleration of rigid body is dependent on multiple parameters such as particle size, impact velocity, sediment and water mechanical properties, and position of the recording sensor with respect to the noise source.

Since the SGN corresponds to continuous random impulses in the river (Geay, 2013), bedload SGN sources cannot be considered scattered point impacts. Instead, bedload SGN sources are here defined as separate areas on the riverbed generating their own acoustic signal. Each area is considered as an independent acoustic source depicting continuously all the noise generated by bedload impacts within the defined area. Hence, the total SGN signal depends on the particle-particle impact signal as well as the number of impacts in each area.

In the presence of multiple acoustic waves emanating from distinct sources, the coherent interaction of these waves transpires through the fundamental principles of superposition and interference, elaborately influenced by the amplitude and phase characteristics of each contributing signal. Notably, the amplitudes are linearly combined, ensuring that their contributions adhere to the principles of linear summation. (Kinsler et al., 1999). When dealing with acoustic energy which is proportional to the square amplitude of the signal, the summation due to different sources will lead to non-linear relationship. However, in this work, we build our method on the assumption of additive effect of the acoustic energy emitted by different impacting particles. The linear addition of acoustic powers can be considered when dealing with random signals in time, such as ambient noise or acoustic emissions from various sources (Vér and Beranek, 2007). In out model, the linearity in adding acoustic energies stems from numerous contributing sources of bedload SGN where each individual source linearly contributes its own energy to the overall acoustic field. This assumption has been widely supported and employed for coherent signal processing and source localization in underwater acoustics (Jensen et al., 2011; Etter, 2018)

The transported bedload is a mixture of sediments impacting the riverbed with different impact rates and intensities depending on the particles diameter, fractional bedload flux, and hydraulic conditions. The riverbed then acts as surfacic acoustic source which emphasizes the spatial distribution of bedload SGN noisce at the surface of the riverbed. In this case, the source power spectral density (PSD, the variation of power with frequency) per unit area $s$ (in $\mu Pa^2 \cdot Hz^{-1} \cdot m^{-2}$) is computed using a linear system that weights the source energy spectrum $e$ ($\mu Pa^2 \cdot s \cdot Hz^{-1}$) generated (at a distance $r = 1$ m) due to impacts of particles of diameter $D_k$ and impact velocity $U_c$ with the corresponding impact rate $\eta$ (number of impacts per second per unit area):

$$s(f, r=1) = \sum_{k=1}^{N_D} \eta(D_k, q_s) \cdot e(f, D_k, r=1, U_c), \quad (1a)$$

$$\eta(D_k, q_s) \propto q_s \cdot \beta(D_k), \quad (1b)$$

where, $N_D$ is the number of classes in the bedload mixture, $q_s$ is the specific bedload flux ($g \cdot s^{-1} \cdot m^{-1}$), $\beta$ is a coefficient dependent on particle saltation trajectory, which is calculated using different empirical equations as a function of particle size, bedload grain-size distribution, and hydraulic conditions (such as water depth and riverbed slope) (Auel et al., 2017; Gimbert et al., 2019; Lamb et al., 2008). Equation (1) shows a linear relation between SGN source $s$ and the specific bedload flux $q_s$ through the impact rate term $\eta$. Then, bedload SGN distribution on the riverbed can be considered as a proxy of the spatial variability of bedload flux in the river cross-section.

## 2.2 Transmission loss function

Acoustic wave propagation refers to the mechanical transmission of the wave and their corresponding energy through a medium. Several processes in rivers are responsible for acoustic waves' attenuation and power losses. The acoustic waves can be attenuated by geometric spreading, refractions or diffractions depending on the geometry of the propagation medium (Geay et al., 2017; Rigby et al., 2016), riverbed roughness (Wren et al., 2015), and riverbed impedance (Etter, 2018). Moreover, the presence of water turbulence and entrained air bubbles induce significant attenuation of acoustic waves (Field et al., 2007).

In shallow water columns such as in rivers, low-frequency acoustic waves are trapped and undergo reflection between the riverbed and the water surface as in a Pekeris waveguide (Pekeris, 1948). In this case, acoustic waves with low frequency are scarcely propagated with a limit frequency called cut-off frequency ($f_{cutoff}$), below which waves don't well propagate (Rigby et al., 2016; Geay et al., 2017). This cut-off frequency is inversely proportional to the riverbed material's, water depth and sound celerity. For example, for a river section with 0.5 m and 2000 m·s⁻¹ as average celerity of sound in sediments (Hamilton,

1987), the cut-off frequency is approximately 1.1 kHz which is lower than bedload SGN frequency range for particles with diameters less than 100 mm (Thorne, 2014).

For frequencies above the cut of frequency, a dimensionless transmission loss function ($TL$) is defined to assess the attenuation of bedload SGN acoustic signal in the river. The $TL$ function depicts the power losses of an acoustic signal propagated from an acoustic source position to any position in the river. Based on experimental work, Geay et al. (2019) proposed that the

propagation function is a combination of a geometrical spreading function $TL_1$, and a frequency-dependent function $TL_2$ that describes the losses of acoustic waves due to the scattering and absorption effects of the river:

$$TL(f,r) = TL_1(r).TL_2(f,r), \quad (2)$$

The function $TL_1$ depicts the decrease of the acoustic power as the waves spread and diverge away from the source. This dimensionless geometrical spreading function reflects the ratio of acoustic intensity (the power per unit area) at a given distance

to the intensity at the source ($r = 1$). For this function, a simplified rectangular geometry of a river section with constant water depth is considered. Depending on the riverbed and water surface interface behavior, two propagation models can be defined. First, if the interfaces act as perfect absorbers (no reflections), the acoustic waves propagate in a spherical mode as in free space (Eq. (3a)). Second, if the interfaces are perfect reflectors, the acoustic waves are trapped between the two interfaces and propagate in a cylindrical way (Eq. (3b)).

$$TL_{1,s}(r) = \frac{1}{r^2}, \quad (3a) \qquad TL_{1,c}(r) = \frac{2}{rd}, \quad (r > h) \quad (3b)$$

where $TL_{1,s}$ and $TL_{1,c}$ are the geometrical spreading functions for spherical and cylindrical models, respectively, $r$ is the source-hydrophone distance (in m), and $d$ is the water depth (in m). The attenuation and losses induced by all other effects and processes, such as water turbulence, are estimated by an exponential propagation function ($TL_2$):

$$TL_2(f) = e^{-2\alpha(f)r}, \quad (4)$$

Where $\alpha(f)$ is a frequency-dependent attenuation coefficient ($m^{-1}$), assumed to vary linearly with the frequency above the cut-off frequency (Jensen et al., 2011) and can be written as follows:

$$\alpha(f) = \alpha_\lambda \cdot \frac{f}{c_f}, \quad (5)$$

where $\alpha_\lambda$ is a dimensionless attenuation coefficient constant characterizing the propagation in the river, with high values corresponding to poor propagation conditions (or higher attenuation of the signal), and $c_f$ celerity of sound in water. Geay et

al. (2019) proposed a protocol for in-situ characterization of $\alpha_\lambda$, which consists in emitting a known calibrated acoustic source (with a loudspeaker) from a fixed point of the river cross-section and measuring the losses of acoustic power per frequency band, with distance. The dimensionless attenuation coefficient can then be fitted using the measurements for both the spherical ($\alpha_{\lambda\,s}$) and cylindrical ($\alpha_{\lambda\,c}$) models. They applied this protocol to seven rivers and concluded that $\alpha_\lambda$ is mainly correlated, positively, with the riverbed slope and roughness. Thus, more attenuation is expected in steep and rough rivers where more

flow turbulence is induced.

The accuracy of acoustic inversion is highly contingent on the precise description of the environment and its corresponding propagation model. In oceanic acoustics, these propagation models have been rigorously investigated and are well-understood, allowing for precise prediction and control of acoustic signals (Roh et al., 2008). Remarkably, the principles of these propagation models bear notable similarity to the seismic wave attenuation phenomena used in seismology (Müller et al., 2010;

Soham and Abhishek, 2016), further demonstrating their validity and utility across different disciplines. For a source in a waveguide, spherical spreading is dominant in the near field. It then transits toward cylindrical spreading when moving away from the source, and cylindrical spreading is dominant in the far field (Jensen et al., 2011). These physical properties have been poorly investigated in rivers, but Geay et al. (2019) showed consistent results with TL calibrated using the spherical and cylindrical model converging at the far field. Because we measure the bedload SGN as close as possible to the noise sources

(see section 4.2), we assume in the following that our acoustic measurements are more dominated by spherical propagation from the near field. This hypothesis is supported by the results of (Nasr et al., 2021), which showed a better performance of the spherical propagation model when compared to the cylindrical one for the majority of the tested rivers.

**2.3 Bedload SGN source**

Consider an acoustic signal generated from a given point source on the riverbed, with a power spectral density $s$ (PSD, in

$\mu Pa^2 \cdot Hz^{-1}$) that propagates to different positions in the river. The signal with PSD $\wp(r)$ (in $\mu Pa^2 \cdot Hz^{-1}$) measured at a distance $r$ from the point source is calculated as the product between the acoustic source spectral power $s$ and the transmission loss function $TL$ (Eq. (6a)). However, in the case of surfacic acoustic sources distributed on the riverbed $\wp$ (in $\mu Pa^2 \cdot Hz^{-1} \cdot m^{-2}$), as defined for SGN, propagation is calculated as a function of area double integral with variable $r$ (Eq.(6b)).

$$\wp(f,r) = s(f) \cdot TL(f,r), \quad (6a)$$

$$\wp(f, x_{hyd}, y_{hyd}, z_{hyd}) = \iint_{x_{s1}, y_{s1}}^{x_{s2}, y_{s2}} s(f, x, y) \cdot TL(f, r(x, y)) dx dy, \quad (6b)$$

where $s(x, y)$ is the source power function which defines the spatial variability of the source in the river, and $r(x, y, z) = \sqrt{(x - x_{hyd})^2 + (y - y_{hyd})^2 + (z - z_{hyd})^2}$ is the distance function between any point on the riverbed with coordinates $(x, y, z)$ and the hydrophone positioned at coordinate $(x_{hyd}, y_{hyd}, z_{hyd})$. The integral limits $(x_{s1}, y_{s1}, x_{s2}$ and $y_{s2})$ define the boundaries of the source in space.

To illustrate the attenuation of acoustic signal due to propagation, Fig.1 presents the acoustic signal for a uniform square unit area acoustic source $s$ ($\mu Pa^2 \cdot Hz^{-1} \cdot m^{-2}$) in addition to the propagated signals with spherical transmission loss function to different distances. This realistic source $s$ was constructed with the Nasr et al. (2021) model for a bedload mixture composed of grains uniformly distributed in the range [1-100 mm], with a specific flux of 1000 g/s/m, and for a river with 1% slope and 1m water level. A value of $\alpha_{\lambda_s} = 0.1$ is used, and two additional values $\alpha_{\lambda_s} = 0.01$ and 0.001 are also considered for r= 2m.

Figure 1b presents the power spectral density PSD (obtained by Fourier transform) of the source $s$ and the propagated signal $\wp$. The losses with increasing distance due to the geometrical transmission loss function $TL_{1,s}$ is evident when comparing the different curves at $r$=2, 5, and 10 m. Simulations at $r$=2m with different $\alpha_{\lambda_s}$ values also illustrate different losses at higher frequencies, captured by the $TL_2$ function (Eq. (5)).

Moreover, we observe a total shift of spectrum to the lower frequencies with distance due to the $TL_2$ function and the increasing

attenuation coefficient with frequency (Eq. (5)). The central frequencies $f_c$ (defined by the condition $\int_0^{fc} |\wp(f)|^2 df = \int_{fc}^{\infty} |\wp(f)|^2 df$) calculated for each power spectrum $\wp$. Between the source position and 10 m, the central frequency decreases from 4.5 kHz to 1.5 kHz. This result illustrates, in particular, how the estimation of transported grain size, which depends mainly on the spectral content, can be misleading without considering the propagation effect.

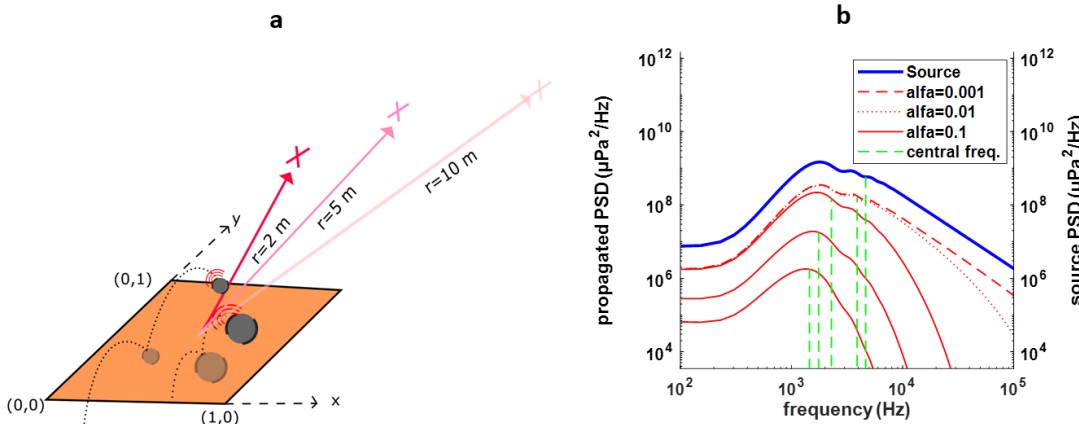

**Figure 1 (a)** representation of a unit surface acoustic source with multiple particle-particle impacts. **(b)** The power spectral density (PSD) of the modelled source signal in blue ($r$=1 m), with the propagated signals in red to $r$=2, 5, and 10 m. Different red line style corresponds to different dimensionless attenuation coefficient for the spherical model ($\alpha_{\lambda_s}$). The green vertical lines represent the central frequency of each PSD.

The physical model of Nasr et al. (2022) calculates the acoustic source of bedload SGN as in Eq. (1) starting from the hydraulic conditions of the river and bedload characteristics (flux and GSD). The latest then modelled the distribution of the propagated SGN in the river ($\wp$) and compared it to measured values. Nasr et al. (2022) concluded that the comparison of the modelled SGN with the measured values is highly dependent on the chosen empirical formula for impact rate ($\eta$) and velocity ($U_c$) (Eq. (1)) which are parameters difficult to validate and measure in the field. In our inversion model, we use the measured SGN ($\wp$) and the transmission loss function ($TL$) to calculate the bedload SGN source ($\mathscr{s}$) which is independent of the propagation characteristics of the river. Equation (1) shows the dependency of the source $\mathscr{s}$ on the bedload flux, however following the results of Nasr et al. (2022) and the limitations on measuring or estimating parameters such as bedload particles impact rate and velocity, the inversion of Eq. (1) to estimate the bedload flux directly from $\mathscr{s}$ will not be covered in this article.

## 3 SGN source inversion method

This section presents the general formulation of the inverse mathematical problem.

### 3.1 Problem formulation

The purpose of the inversion problem is to estimate the PSD and the spatial distribution of bedload SGN sources in rivers. The problem can be illustrated in Figure 2, where $M$ bedload SGN sources of constant width $W_M$ are assumed to be distributed on the riverbed with total width $W$. It is assumed that the specific bedload flux ($\bar{q}_{s,x}$) is constant for each band in the streamwise direction (along longitudinal line y); in other words, source power is assumed uniform along a given longitudinal line. This simplifies the geometry of sources as planar strips with infinite lengths in the y direction (Figure 2). The PSD per unit area $\mathscr{s}_m(f)$ ($\mu Pa^2 \cdot Hz^{-1} \cdot m^{-2}$) is defined for each source with $m$ an integer $1 \leq m \leq M$. The vector $\boldsymbol{S}$ of dimension $[M,1]$ and with the elements $\mathscr{s}_m(f)$ represent all the sources' PSD distributed in the river.

To solve the inversion problem, the first parameter to be considered is the PSD of acoustic measurements of the bedload SGN. Here, we consider a situation matching with drift boat measurement, where a boat supporting the hydrophone successively measures the associated acoustic SGN at $N$ different positions. Measuring the SGN noise using a freely drifted boat with the flow significantly reduces the hydraulic noise generated by hydrophone resistance to the flow (Geay et al., 2020). $N$ acoustic measurements are thus assumed to be distributed on the river cross-section (x direction Figure 2), from which we compute a PSD for each drift measurement. The parameter $\wp_n(f)$ corresponds to the PSD measured by a hydrophone drift at the n$^{th}$ position with $n$ an integer $1 \leq n \leq N$. The measured SGN profile is thus represented by the vector $\boldsymbol{\mathcal{P}}$ with dimension $[N,1]$ comprising all measured $\wp_n(f)$.

Given all sources in the river, the measured PSD $\wp_n(f)$ is the contribution of all $M$ acoustic sources propagated to the n$^{th}$ measuring position. The contribution of all propagated source signals to the measured PSD can be calculated using the linear equation as follow:

$$p_n(f) = \sum_{m=1}^{M} a_{m,n}(f) \cdot s_m(f), \quad (7)$$

where $a_{m,n}$ is the attenuation factor that affects the propagated signal of source $m$ when measured by the hydrophone at position $n$. The attenuation factor $a_{m,n}$ is calculated for a surfacic source using the frequency-dependent transmission loss function $TL$:

$$a_{m,n} = \int\!\!\int_{x_{m1},y_{m1}}^{x_{m2},y_{m2}} TL\big(f(k), r_{m,n}(x,y,z)\big) dx\, dy \quad (8)$$

Where $r_{m,n}$, is the function that defines the distance between any point within the source $m$ area and a hydrophone at position $n$ with coordinate $(x_{hyd,n}, y_{hyd,n}, z_{hyd,n})$ such that $r_{m,n}(x,y,z) = \sqrt{\left(x - x_{hyd,n}\right)^2 + \left(y - y_{hyd,n}\right)^2 + \left(z - z_{hyd,n}\right)^2}$, and $z$ depends on the geometry of the section (constant for rectangular cross-section). The integral limits $x_{m1}, y_{m1}, x_{m2}$ and $y_{s2}$ define the boundaries of the source in space. The values of $y_{m1}$ and $y_{m2}$ were chosen to be much greater than the river width $W$ (length $= 10W$) to model the infinite length of the source stripe. Finally, when Eq. (7) is applied to the whole domain we obtain the matrix:

$$\boldsymbol{P} = \boldsymbol{\mathcal{A}} \cdot \boldsymbol{S} \quad (9a)$$

$$\begin{pmatrix} p_1(f) \\ \vdots \\ p_N(f) \end{pmatrix} = \begin{pmatrix} a_{1,1}(f) & \cdots & a_{M,1}(f) \\ \vdots & \ddots & \vdots \\ a_{1,N}(f) & \cdots & a_{M,N}(f) \end{pmatrix} \cdot \begin{pmatrix} s_1(f) \\ \vdots \\ s_M(f) \end{pmatrix} \quad (9b)$$

where $\boldsymbol{\mathcal{A}}$ is the attenuation. The multiplication of the $n^{th}$ raw elements of attenuation matrix $\boldsymbol{\mathcal{A}}$ with the sources vector $\boldsymbol{S}$ corresponds to the propagation of all sources in the river to the $n^{th}$ hydrophone position.

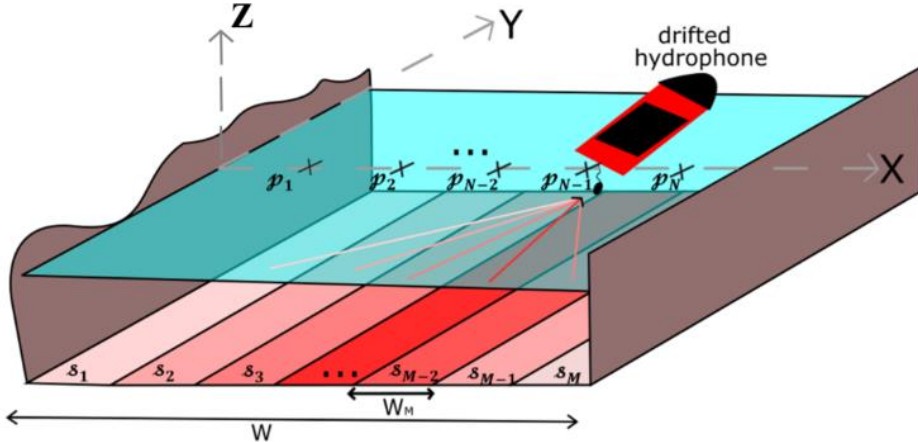

**Figure 2** Comprehensive presentation of the inversion problem geometry where $s_m$ corresponds to different bedload SGN sources on the riverbed. The difference in color corresponds to different SGN source intensities. The points $p_n$ corresponds to SGN measurements at different positions by the drifted hydrophone method.

## 3.2 Solution to the inversion problem

At this stage, we consider that we know the measured acoustic matrix $\mathcal{P}$ and assume that the attenuation matrix $\mathcal{A}$ is computed (Eq. (3) and (4)) with a known (measured) attenuation term $\alpha_\lambda$. We seek the solution $\widehat{S}$ of the vector $S$, which allows the modelled vector $\widehat{\mathcal{P}} = \mathcal{A} \cdot \widehat{S}$ to best fit the measured acoustic $\mathcal{P}$ vector. A traditional approach for this type of problem is the Least Square (LS) method, with an optimization algorithm that works on the minimization of squared residual errors between $\mathcal{P}$ and $\widehat{\mathcal{P}}$. The error vector $\epsilon$ can be written as in Eq. (10a), and the optimization of the problem solving $\widehat{S}$ is presented in Eq.

(10b), where the argument of the minimum of $\epsilon$ ($\arg\min(\epsilon)$) is the value of $\widehat{S}$ that minimize $\epsilon$.

$$\epsilon = \mathcal{P} - \widehat{\mathcal{P}} = \mathcal{P} - \mathcal{A} \cdot \widehat{S} \quad \text{(10a)}, \qquad \widehat{S} = \arg\min(\epsilon) \quad \text{(10b)}$$

The relation between the number of sources $M$ and measurements $N$ determines the type of algebraic for the problem in Eq. (9). If the number of sources exceeds the number of measurements ($M > N$), then the equation is considered under-determined. In this case, there are more unknowns than equations and an infinite number of solutions of $\widehat{S}$ exists. On the other hand, if

$M < N$, there are more independent equations than unknowns, and the equation system is considered over-determined. In the latest case, it is shown by (Nelson and Yoon, 2000) that the optimal solution for the acoustic source vector, which ensures minimization of Eq. (10b), is :

$$\widehat{S} = \mathcal{A}^+ \cdot \mathcal{P} \quad \text{(11)}$$

where $\mathcal{A}^+ = (\mathcal{A}^t \cdot \mathcal{A})^{-1} \cdot \mathcal{A}^t$ is the pseudo-inverse of the matrix $\mathcal{A}$ and $\mathcal{A}^t$ is the transpose matrix.

The pseudo-inverse algorithm for non-square matrixes exhibits a common drawback where the solution $\widehat{S}$ may suffer from divergence (instability) under slight variations in the value of the elements of $\mathcal{A}$ or $\mathcal{P}$. The problem's ability to estimate stable or unstable solution $\widehat{S}$ is called conditioning of the problem. The conditioning of the problem is quantified by the condition number $\sigma$ of the matrix $\mathcal{A}$ to be inversed. This condition number is defined as $\sigma(f) = \|\mathcal{A}\| \cdot \|\mathcal{A}^-\|$ where $\|\mathcal{A}\|$ is the 2-norm of the matrix $\mathcal{A}$ (Golub and Van Loan, 1996). A system with a high value of $\sigma$ is considered an ill-conditioned system that

generates high instability of the solution $\widehat{S}$ to small deviation or error in $\mathcal{A}$ and $\mathcal{P}$. In contrast, a value $\sigma$ closer to 1 is a well-conditioned system. A problem with a condition number $\sigma < 10^3$ can be considered a well-conditioned (Arthur et al., 2017).

In addition, relatively high resolution of hydrophone measurements ($N >> M$, or close measurements) will lead to matrix $\mathcal{A}$ with the close values of attenuation factor ($a_{m,n}$) at the same row, consequently, rank deficient matrix. A classical solution for such instability problems is the non-negative least square (NNLS) method, a constrained least squares problem where the

values in the solution vector $\widehat{S}$ are strictly positive values.

In the case of the number of sources equal to the number of measuring points ($N = M$), then the pseudo-inverse matrix is simply the algebraic inverse matrix of $\mathcal{A}$ and $\widehat{S} = \mathcal{A}^- \cdot \mathcal{P}$.

The Matlab function *lsqnonneg ()*, which follows the NNLS algorithm, is used for solving the inversion problem.

### 3.3 Numerical testing of the inversion model

Several numerical tests are presented here to illustrate the behavior and limits of the proposed inversion model. The tested section is composed of a 10 m wide river, with a rectangular section and a 1-meter water depth. Bedload SGN sources are assumed to be distributed on the riverbed in the form of bands, as in Figure 2. The total bedload active channel width —the sections with bedload transport—equals 4 m. Within the active bedload channel, the source PSD $s_m$ is computed with Nasr et al. (2021); outside $s_m$ is zero. Three different configurations of bedload transport distribution have been tested (single, dual,

and triple channels) which correspond to the number of separated bedload active channels in the river cross-section (Figure 3). The considered length of the sources along the river direction is 100 m upstream and 100 m downstream of the section.

We consider the number of simulated acoustic measurements equal to the number of sources ($M = N$), and the measurements are positioned above each source's center (Figure 2). The simulated PSD $p_n$ are calculated using the PSD of the acoustic sources $s_m$ as in Eq. (7). The spherical propagation model is used with an attenuation coefficient $\alpha_{\lambda_s} = 0.05$ (equivalent to

propagation environment for a river with slope $S \approx 1\%$).

Figure 3 shows the cross-sectional distribution of the frequency-integrated source power $P_{s_m}$ ($\mu Pa^2 \cdot m^{-2}$, in blue line) and simulated measured power $P_{p_n}$($\mu Pa^2$, in red line) for different configurations, such that:

$$P_{s_m} = \int_{f_{min}}^{f_{max}} s_m(f)\, df, \quad (12a) \qquad P_{p_n} = \int_{f_{min}}^{f_{max}} p_n(f)\, df, \quad (12b)$$

In the absence of hydraulic noise at low frequencies (Geay, 2013), $f_{min} = 0\ kHz$ and $f_{max} = 150\ kHz$, which is the maximum

value of the simulated PSD.

In the first place, no extrinsic acoustic noise has been considered. Using the simulated acoustic profile $\mathcal{P}$ and Eq. (11), the sources PSD are inversed by NNLS method for different tests. Figure 3 shows that the inversed source power profiles $P_{\widehat{s_m}}$ (in black line) coincide with generated profile (in blue) for all tests suggesting good prediction and solution of NNLS under accurate measuring conditions.

To account for possible uncertainty in field measurements, a noise has been added to the simulated $p_n$. The noise was added in the form white noise signal convolved with the SGN signal. The resulting acoustic profiles are plotted (dashed red lines) in Figure 3. In the presence of noise, the inversed source power $P_{\widehat{s_m}}$ (dashed black lines) is distinct from the generated source power profile (in blue). The results errors are not only limited to the intensity of sources but also the appearance of sources outside the bedload active channel. Nonetheless, the average cross-sectional power of the inversed source profile (integration

of the curve divided by the width) is between 2.35-2.43 $\mu Pa^2$/m, which is close to the corresponding value for that for the imposed source (2.36 $\mu Pa^2$/m). This means that if we consider the total inversed power, the error is more limited to the localization of these sources.

To numerically assess the results, a variance-explained accuracy measure ($VEcv$) parameter is introduced (Li, 2017). The advantage of this dimensionless accuracy measure $VEcv$ that it is independent from data mean, and variance according to its

definition. A $VEcv$ close to one means good accuracy of the model. The $VEcv$ us calculated as follow:

$$VEcv = \left(1 - \frac{\sum_{m=1}^{M}\left(P_{s_m} - P_{\widehat{s_m}}\right)^2}{\sum_{m=1}^{M}\left(P_{s_m} - \bar{P}_s\right)^2}\right), \quad (13)$$

where, $P_{\widehat{s_m}}$ is the inversed source power, $P_{s_m}$ is the imposed source power, $\bar{P}_s$ is the average of all imposed source power. The values of $VEcv$ have been calculated for each simulation and are presented in the titles of Figure 3. The $VEcv$ values show that the inversion model can have good performance even in the presence of noise ($VEcv \approx 0.9$ close to 1). However, the $VEcv$ values relatively decrease when the number of bedload active channels increases, suggesting a higher sensitivity of the model to field uncertainty under complex bedload distribution.

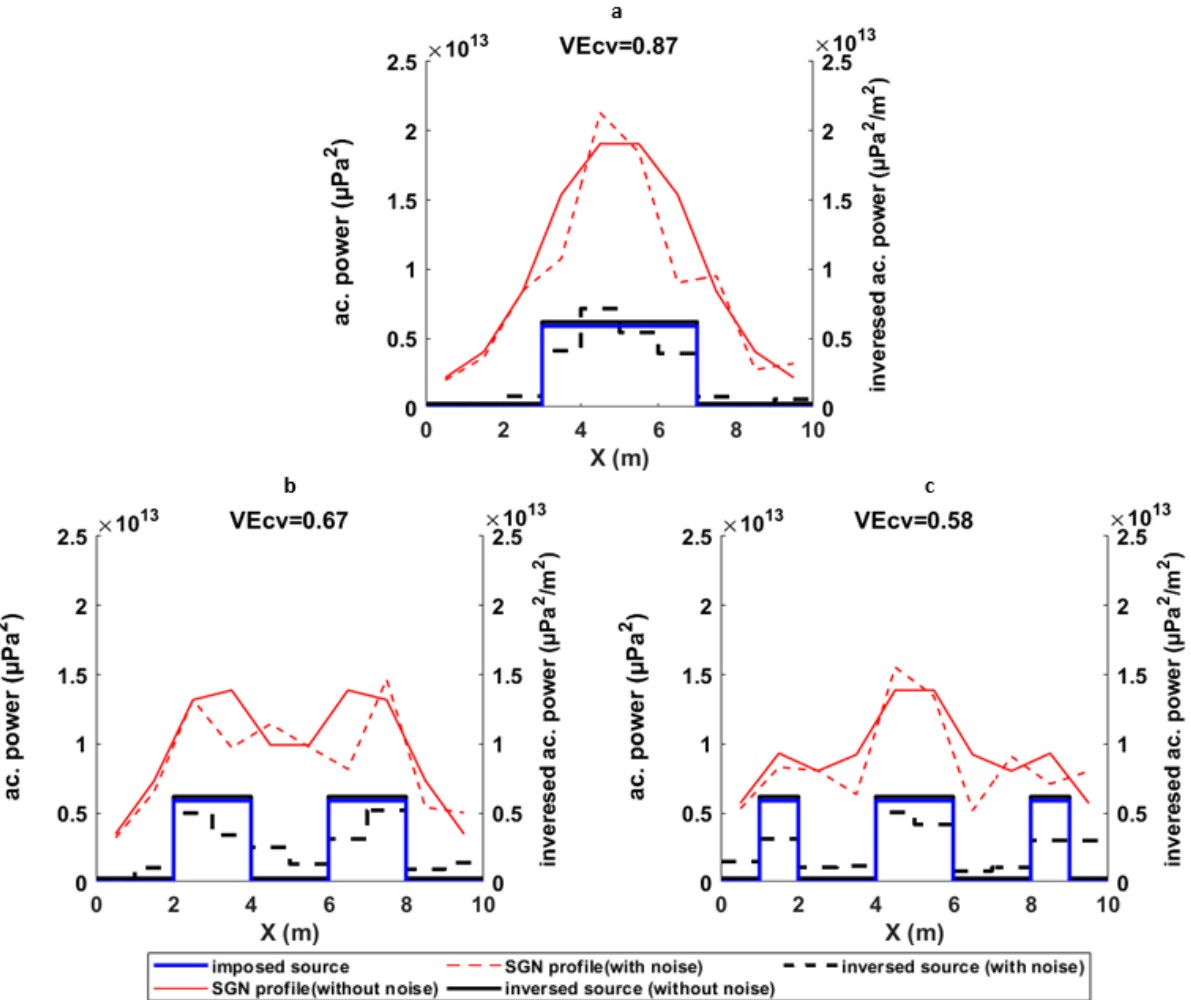

**Figure 3** Numerical test results of the inversion model for (a) single, (b) double, (c) triple bedload active width channel configuration. The figures compare the simulated SGN source acoustic power $P_{s_m}$ (in blue, $\mathbf{\mu Pa^2 \cdot m^{-2}}$ ) with the inversed source power $P_{\widehat{s_m}}$ (in black) without noise (continuous line) and with noise (dashed line). The figure also compares the measured SGN acoustic power $P_{p_n}$ (in red, $\mathbf{\mu Pa^2}$) without noise (continuous line) and with noise (dashed line).

## 4 Inversion model validation

In this section, we will present two experiments for testing and validating the inversion model.

### 4.1 Validation with active test measurements

This first experiment aims to test the inversion model under controlled source conditions. It is technically challenging to deploy a sound source with a scale comparable to the SGN source in the river. Instead, in this experiment, we use a loudspeaker in the river as a source with a known signal and location. The test consists of measuring the emitted sound by the loudspeaker at different locations in the river and then testing the ability of the inversion model to retrieve the active source's location and PSD.

### 4.1.1 Isere River and experimental setup

This experiment was carried out in the Isere River in south-eastern France. The measuring site is located next to Grenoble city (45°11'55. .0"N 5°46'11.4"E) on a pedestrian's footbridge crossing the river. The local average slope for the measured section is 0.05% with 60 m width, and the annual average flow is 180 $m^3 \cdot s^{-1}$. The riverbed is composed of gravel with average $D_{50,bed} = 23$ mm measured with the Wolman (1954) sampling protocol for the exposed riverbed. During the time of measurement, on the 25$^{th}$ of August 2022, the average flow was 110 $m^3 \cdot s^{-1}$. Under this flow condition, the Isere River is characterized by low hydraulic noises generated by the flow turbulence at low frequencies (Geay, 2013), as well as no bedload SGN.

We used a waterproof piezoelectric loudspeaker Lubell with a 23 cm diameter (model LL916H; http://www.lubell.com/LL916.html), characterized by a quasi-flat (+/-10dB) frequency response between 500Hz-21,000Hz. The loudspeaker is connected to an emission RTSys system (TR-SDA14) which controls the emitted signal by a ".wav" file stored inside the RTSys. The chosen transmission signal is a logarithmic frequency modulation between 500 Hz and 21 000 Hz in 0.25 seconds. The ".wav" file for the sound emitted by the loudspeaker is provided in the supplementary material. The loudspeaker signal was characterized in a lake next to Grenoble city in France. The water depth at the testing position was around 5.5 m. The source was positioned 3 m under the water's surface. The emitted signals were measured at a 1 m horizontal distance from the source with an HTI-99 hydrophone (High Tech, Inc., http://www.hightechincusa.com), with a sensitivity of -199.8 dB and characterized by a flat frequency response ($\mp$ 3dB) between 2 Hz and 125 kHz. The hydrophone was connected to the EA-SDA14 card acquisition system (RTSYS company) recording the acoustic signal in ".wav" format with a sampling frequency of 312 kHz. Different orientations of the loudspeaker in space have been tested. A PSD ($\mu Pa^2 \cdot Hz^{-1}$) was calculated for each measured chirp. Finally, using Eq. (6b), the surfacic PSD of the loudspeaker ($\mu Pa^2 \cdot Hz^{-1} \cdot m^{-2}$) was calculated by dividing the measured PSD with the $TL$ function term. The $TL$ function was calculated considering the dimension of the source

for $r = 1$ m, and $\alpha_{\lambda_s} = 0$, attenuation being only due to geometrical spreading in a lake. The result of the source surfacic source power is presented in Figure 4c (green lines) with the 5%, 50% 95 % percentiles.

In the Isere River, the loudspeaker has been deployed from the bridge to the riverbed at the position $x_{source} = 48$ and $y_{source} = 3$ m (in the downstream direction). At this position, the average water column depth is 1.5 m. The signals were emitted from the source in an endless loop. We measured the acoustic profiles every 2 m between $x = 8$ and $x = 56$, with the same hydrophone and acquisition system presented above. The protocol was identical to Geay et al. (2020), with the hydrophone mounted on a floating river board (40 cm below the water surface), and freely drifted from the bridge (drift position

between $y = 2$ m and $y = 4$ m from the bridge). The acoustic measurements have been carried out on $N$ different position on the river cross-section. For each drift $n$ located at $x_n$, we measured the power spectrum of all signals impulsion during the drift and determined the median spectrum $PSD_n$. Each drift $n$ is now characterized by its coordinate $(x_n, y_n = 3$ m), and a median spectrum $PSD_n$.

Inversion of the active acoustic source requires the definition of parameters presented in Eq. (9) ( $\mathcal{P}$, $\mathcal{S}$ vectors and $\mathcal{A}$ matrix).

For the measured acoustic profile, the vector $\mathcal{P}$ is composed of the 25 measured median power spectrum defined above, $p_n(f) = PSD_n(f)$ (1< $n$ <25). We considered that $N = M = 25$ and incorporate 25 square sources of 2-meter side distributed between $7 \leq x \leq 56$, with unknown source power spectrums $s_m(f)$. The transmission loss parameters $a_{m,n}(f)$ have been calculated using Eq. (8) for the spherical model. The attenuation coefficients presented in Eq. (4),      $\alpha_{\lambda_s} = 10^{-4}$ have been estimated following the protocol proposed by (Geay et al., 2019) during the measurements day. To reduce the computational

load the sources spectrum $s_m(f)$ have been calculated using the third-octave band of the measured spectrum.

The area of the inversed sources in this application is 4 m$^2$ (2 m side squares) which is different than that of the loudspeaker area $\approx 0.04$ m$^2$. In this case, an area correction factor was applied to the inversed results in order to compare it with the loudspeaker source signal measured in the lake. The area correction factor was calculated as the ratio between the TL function calculated as in Eq. (6b) for the inversed source area and for the loudspeaker area.

### 4.1.2 Results


Figure 4b plots the measured acoustic power profile $P_{p_n}$ (in red line), calculated with Eq. (12) between frequencies of 500-21000 Hz. The measured spectrums show different intensities depending on the distance from the active source. No significant variation in the spectral distribution is observed with propagation due to the relatively low attenuation coefficient in the Isere River.

The results of the inversed power profile ($P_{\widehat{s_m}}$) are plotted in Figure 4b (black line). The results show that the inversion model successfully captures the active source location between $x = 47$ and $x = 49$ ($m = 21$). However, some residual sources have been modelled mainly around the active source location and at other locations in the river. As in the numerical test with noise (section 3.3), it is suspectable that measurement uncertainty contributes to such residual sources as they coincide with the perturbation in the measured acoustic profile (e.g., $x = 26$ and 35 m).

The spectrum of each drift ($\mathscr{p}_n(f)$) are presented in Figure 4c (continuous faded lines), and the color index corresponds to the distance of the spectrum from the deployed loudspeaker. Figure 4c shows the inversed source spectrum in the proximity of the loudspeaker location $\hat{s}_{n=21}$ (between $x = 47$ and $x = 49$). The results show that the inversed spectrums are comparable with the reference spectrum of the source characterized in the lake, which fits within the 5%-95% percentiles on most frequencies.

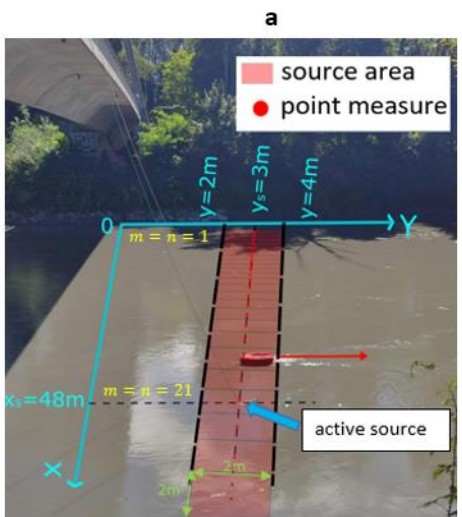

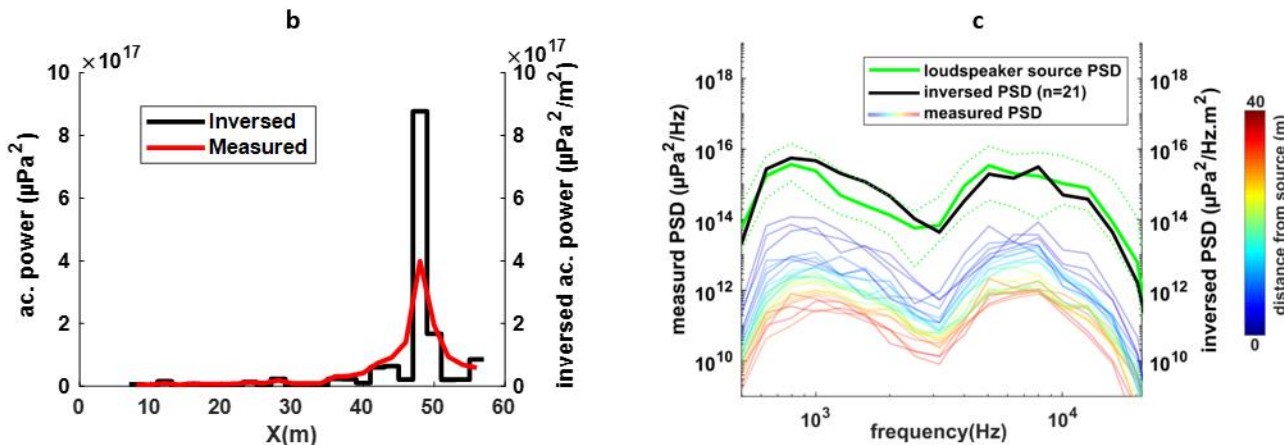

**Figure 4 (a)** Representation of the geometry of sources and acoustic measurements on the Isere River. **(b)** measured acoustic power $P_{\mathscr{p}_n}$ ($\mathbf{\mu Pa^2}$, red line) and the inversed source power $P_{\widehat{s}_m}$ ($\mathbf{\mu Pa^2 \cdot m^{-2}}$, black line). **(c)** measured spectrum at each position in the river $\mathscr{p}_n$ ($\mathbf{\mu Pa^2 \cdot Hz^{-1}}$, faded color lines). The color index shows the distance of the measurement from the source which increases from blue to red. The inversed spectrum $\widehat{s}_m$ ($\mathbf{\mu Pa^2 \cdot Hz^{-1} \cdot m^{-2}}$, in black line) corresponds to the spectrum at location 47-49 m (*n*=21). The green spectrum corresponds to the median of the lake measured spectrum with the dotted line corresponds to 5% and 95% percentiles.


## 4.2 Validation with passive SGN measurements

### 4.2.1 Giffre River and experimental setup

In this part, we apply the inversion model to bedload SGN measurements. An experiment was carried out in the Giffre River located in French Alps. The measured section is under a pedestrian crossing bridge (46°04'48.8"N 6°42'19.4"E). The average slope of the section is 0.3% and 29 m in width. Two measurements of SGN and bedload flux were carried out during the melting season on 13$^{th}$ of June 2018 and 6$^{th}$ of July 2021. On these dates, the flow discharges were 50 m$^3$/s and 26 m$^3$/s respectively with 0.9 m and 0.7 m average water depth ($d$).

Acoustic measurements were obtained using HTI-99 hydrophones (with sensitivity: -200.1 dB in 2018 and -199.8 dB in 2021) and RTSys acquisition system with the drift protocol (Geay et al., 2020). The drifts were 20 to 30 meters long (in the y direction) with the hydrophone setup 30 cm below the surface. Several repetitions of drifts have been performed at each cross-sectional position $x_n$ to account for measurement uncertainty and temporal variability. For each drift at the location $x_n$, we computed the median measured $PSD_n(f)$. In the presence of repetition of drifts at the same location $x_n$, we averaged the $PSD_n$. Bedload particles were sampled from the bridge using a handheld Elwha sampler of dimensions 203 × 152 mm. Sampling was performed at various cross-section positions following the procedures proposed by (Edwards and Glysson, G, 1999) with variable repetitions. Each sample was dried, sieved, and weighed to calculate the transport rate and grain size distribution (GSD). We calculated a specific bedload flux $q_{s,i}$ (in g · s$^{-1}$ · m$^{-1}$) as follows:

$$q_{s,i} = \frac{m_i}{t_i \times W_{\text{sampler}}}, \quad (15)$$

where $W_{\text{sampler}}$ is the inlet width of the sampler, $m_i$ and $t_i$ are the mass and the duration of sampling respectively. The average bedload flux profile has been calculated within $N$ windows, each of 2 meters in width. Each window is centered on an acoustic point measurement $x_n$ as for the acoustic source. The average bedload flux $\bar{q}_{s,n}$ (in g · s$^{-1}$ · m$^{-1}$) for the window $n$ is calculated by averaging the values of $q_{s,i}$ contained inside the spatial window $n$.

The geometry of the SGN sources used is similar to Figure 2 with a length extended for each source between $y$ =-150 m and $y$ =150 m which account for the infinite length assumption of the SGN sources.

Two active tests following the protocol of Geay et al. (2019) have been carried out to characterize the propagation environment in the Giffre River during the two measurement days in 2018 and 2021. The attenuation coefficients estimated for the spherical model are $\alpha_{\lambda_s} = 0.006$ and $\alpha_{\lambda_s} = 0.004$ for 2018 and 2021 respectively. The attenuation coefficients were measured up to a maximum frequency of 20 kHz and extrapolated at higher frequencies assuming a linear regression.

### 4.2.2 Results

Figure 5 presents the averaged profile for bedload flux and the measured acoustic power for both experiments. The left and right riverbanks are located $at\ x = 0$ and $x = 29$, respectively. In both measurements, the bedload flux profile is composed of a main transported channel localized at the right section of the river (peak at $x = 20$). The average specific bedload flux

calculated for both experiments shows that the bedload transport intensity in 2018 was fifteen times more than that of 2021 ($328 \text{ g} \cdot \text{s}^{-1} \cdot \text{m}^{-1}$ compared to $22 \text{ g} \cdot \text{s}^{-1} \cdot \text{m}^{-1}$). The measured SGN profiles show a coherent variation of acoustic power with the bedload flux in the river cross-section. However, the decrease of the acoustic power in the left part of the river section (between $x = 0$ and $x = 13$) does not correspond to the same intensity decrease of bedload flux.

Acoustic recording samples from both experiments are presented in the supplementary materials. After analyzing and listing the recordings at different frequencies, bedload SGN can be clearly heard above 800 Hz. At frequencies lower than 400 Hz, the main source of noise is the hydraulic noise induced by the flow turbulence in the river and around the hydrophone. The mean measured PSD are presented in Figure 6a and 6b. The central frequencies calculated for the mean PSD are 5.6 kHz and 10 kHz for 2018 and 2021 respectively. The difference in central frequency is mainly induced by the different grain-size distributions sampled during both experiments (the average $D_{50}$ sampled in 2018 was 6.8 mm and 3 mm in 2021). In addition, the attenuation of the SGN signal is more important during 2018 measurements due to more water turbulence induced by the higher flow. The higher attenuation contributes to the decrease in the measured central frequency as explained in section 2.3.

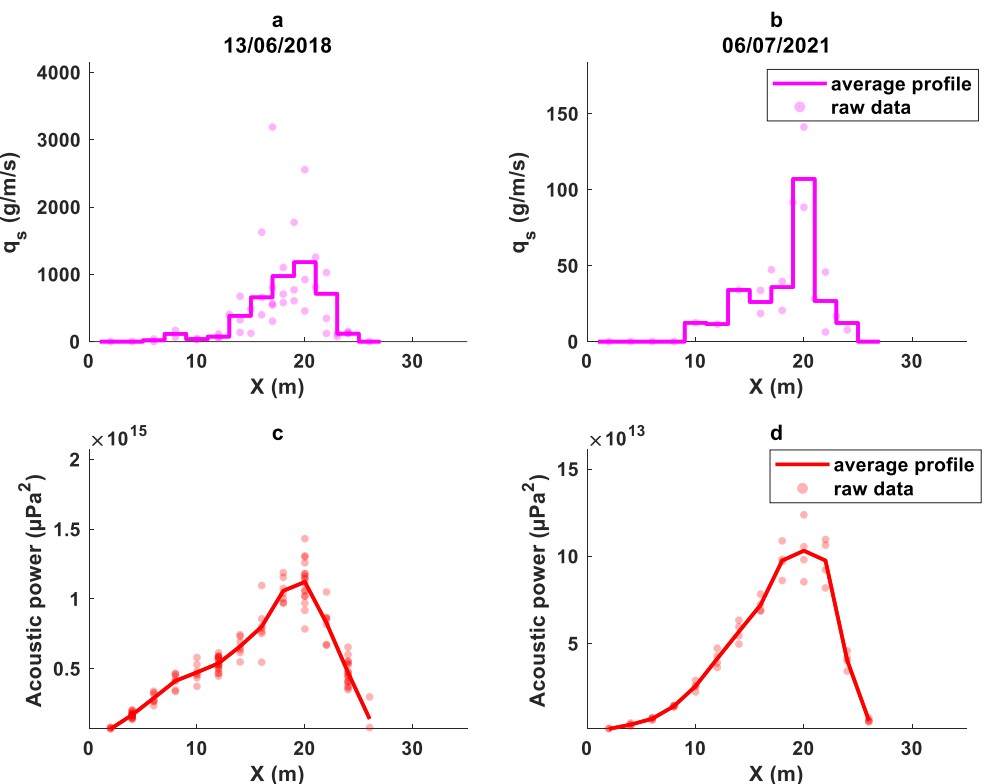

**Figure 5** Measured bedload flux **(a)** in 2018 experiment and **(b)** in 2021 experiment. The measured acoustic power **(c)** in 2018 experiment **(d)** in 2021 experiment.

Figures 6a and 6b also present the mean inversed PSD. The central frequency calculated for the inversed PSD shows an increase in both experiments compared to the measured value (an increase from 5.6 kHz to 11 kHz in 2018 and 10 to 19.1 kHz in 2021). A visual comparison shows that other than the power value, the main difference is the slope of the PSD at higher

frequencies. In contrast, the PSD shape at lower frequencies has not been significantly affected. This shows that the inversion model corrects the attenuated signal at high frequencies, as explained in section 2.3 and Figure 1.

Figures 6c and 6d present the inversed power profile $P_{\widehat{s_m}}(\mu Pa^2 \cdot m^{-2})$ which can be compared to the measured profile $P_{p_n}$. The inversed power per unit area is one order of magnitude less than the measured power since each source contributes (by sound propagation and in a cumulative way) to each measured value. Moreover, the source spectrum was calculated for a

distance of 1m from the source, while the measurements with the drift hydrophone were taken at a smaller distance (~ 30cm below the water surface <1 m).

To compare the measured and inversed power with the bedload flux profile, we scaled the signals by computing the ratio between the local value and the total cross-sectional value for each profile. Results are plotted in Figures 6e and 6f, which show a better synchronization of the bedload flux profile with the inversed power profile than the measured profile. This is

particularly evident when considering the peaks and the sharp transition to low transport at the side of the section.

To numerically compare the profiles, the $VEcv$ value is calculated for both the relative source and the relative measured profile in reference to the relative bedload flux profile. The values of $VEcv$ are presented in Figures 6e and 6f, confirming that the inversed source profile better illustrates the bedload flux than the measured SGN profile in both experiments. However, the improvement of $VEcv$ is with less extent in the 2021 experiment than that for the 2018 experiment.

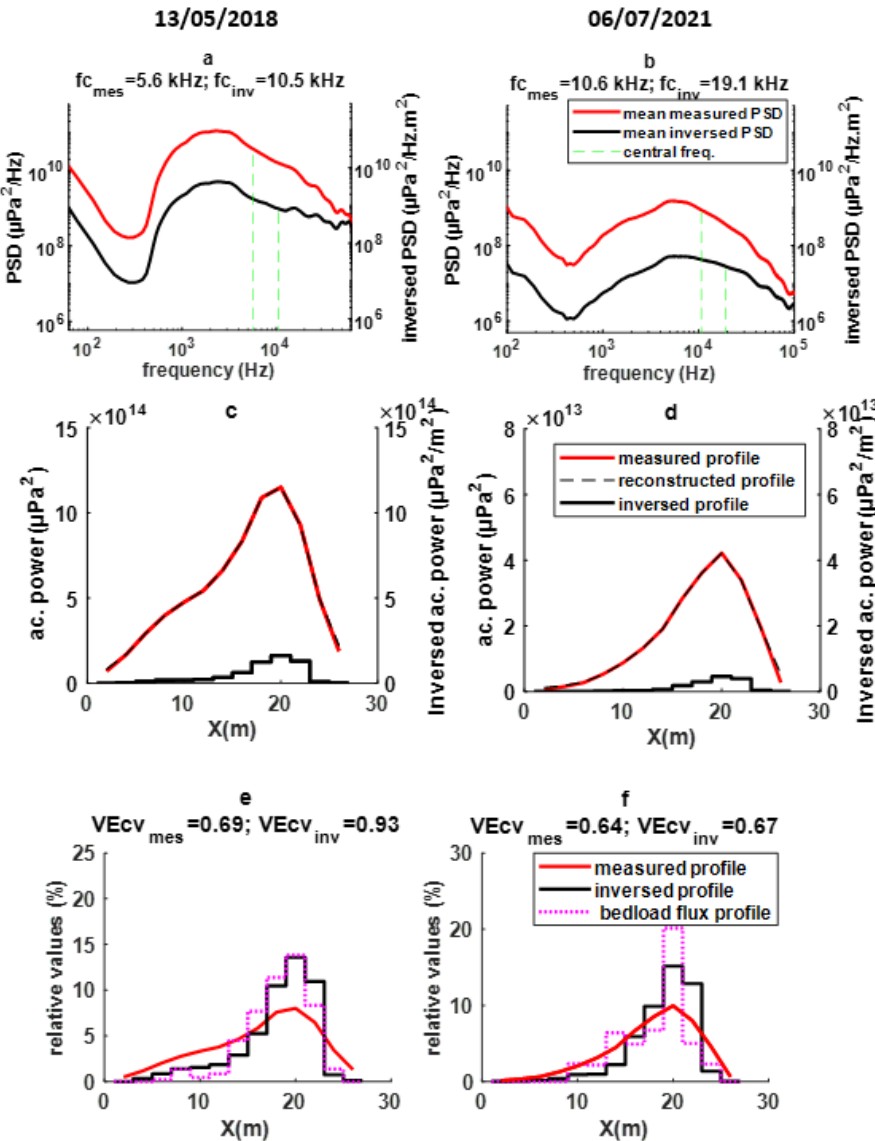


**Figure 6** Mean measured PSD $\boldsymbol{p_n}$ (in red) and inversed PSD $\widehat{\boldsymbol{s_m}}$ (in black) for 2018 and 2021 in **(a)** 2018 and **(b)** 2021. Mean measured power $\boldsymbol{P_{p_n}}$ (in red) and inversed power $\boldsymbol{P_{\widehat{s_m}}}$ (in black) for c) 2018 and d) 2021 experiments. Relative profiles for **(e)** 2018 and **(f)** 2021, in magenta is the relative bedload flux profile $\overline{\boldsymbol{q}}_{s,n}$.

To study the effect of inversion on the acoustic power-bedload flux relation, the measured bedload flux value at measuring position $n$ is plotted against the corresponding value of measured acoustic power and inversed acoustic power for both 2018 and 2021 experiments (Figure 7). Depending on the experiment, we can differentiate two different trends for the measured acoustic power.

    Power laws have been fitted in Figure 7 to the measured data by applying reduced major axis regression RMA which is used

when data on both axes have uncertainties (Smith, 2009). The fitted power laws presented in Figure 7 show two very distinct trends, with more than one order of magnitude of bedload flux for the same acoustic power values. On the other hand, the relationships obtained with the inversed data show a better continuation with less dispersion between the two experiments, allowing a unique fit with a relatively good Pearson correlation coefficient ($R^2$=0.79).

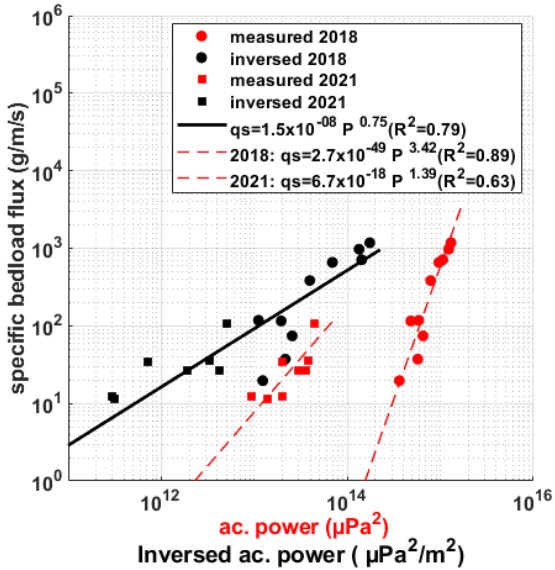


**Figure 7** Local values of bedload flux vs local values of measured acoustic power (in red) and inversed acoustic power (in black)

## 6. Discussion

### 6.1 Dealing with uncertainties

    The numerical testing in section 3.3 (Figure 3) showed that the comparison between the simulated and the inversed source profile is impacted by the presence of acoustic noise in the signal. In addition, the Isere experiments results (Figure 4b) have

shown that extraneous noise sources appeared in different positions with different intensities due to uncertainty and perturbations in the measured acoustic profile. Meanwhile, the data collected on the Giffre river shows variabilities in acoustic

measurements as well as the bedload flux measurements (Figure 5). Thus, the inversion results of the Giffre application should consider the potential errors due to measuring uncertainties. These uncertainties have been calculated following Geay et al. (2020), who estimated the relative uncertainty of acoustic measurements at 8% and 6%, and the relative uncertainties associated to bedload flux at 29% and 32%, for 2018 and 2021 respectively. Several factors can contribute to the variability of bedload flux measurements, such as the efficiency of bedload samplers itself under different hydraulic conditions (Childers, 1999; Bunte et al., 2008). Moreover, the uncertainty of bedload sampling is also affected by the position of the sampler on the river bed (Vericat et al., 2006), where difficulties in controlling the exact position of the Elwha sampler were reported during our field measurements. Besides, in the 2021 experiment, the number of bedload sampling repetitions was limited due to unstable flow condition generated by a rainfall event after the beginning of the experiment. Then, the main difficulty in comparing inversed acoustic measurements with bedload flux profile is mainly related to the quality of bedload flux sampling. Additional uncertainties also concern the attenuation coefficients obtained by fitting measurements of active test data showing variability up to a factor of 5 around the best fit.

## 6.2 Improvement of the calibration curve

The hydrophone measures not only its close environment but all sounds propagating in the river section. The cross-section integration results depend on the local condition, which can change with discharge, as shown in Figure 6c and 6d. This explains the two different fits between bedload flux and acoustic power obtained for the Giffre river in Figure 7. In addition, the high-power coefficients (1.4- 3.4), greater than unity as predicted by the theory (Nasr et al. 2022), are also a consequence of the overestimation of the actual source energy. The global calibration curve of bedload flux obtained by is based on the average cross-sectional acoustic power values. The effects discussed here have probably contributed to part of the variability obtained when they fit bedload flux as a function of acoustic power. More importantly, the global calibration curve may also generate an overestimation of bedload flux under certain conditions. For example, this calibration curve has been tested on the Drac River (a tributary of the Isere River), which is characterized by good sound propagation of SGN and a well-localized bedload channel. The result was an overestimation of the annual average bedload flux by a factor of more than 3.

Figure 7 shows that reducing these effects by inversing the acoustic power gives access to a better adjustment of the data obtained under different bedload transport conditions between 2018 and 2021 experiments. This offers a good potential for improving the global calibration curve (Geay et al., 2020; Nasr et al., 2023), by adjusting a new function after inversing their whole data set. We used the inversion model on the data set of the global calibration curve presented by Nasr et al. (2023), which consists of 42 experiments of simultaneous bedload flux and acoustic measurements collected in 14 different rivers, covering a wide range of properties (e.g., slope, bedload intensity, and granulometry). The inversion model has been applied to all rivers, similarly to the Giffre river application. In the case of the absence of an active test on some rivers, a slope-based empirical formula derived from field data (Geay et al., 2019; Nasr, 2023) has been used to estimate the attenuation coefficient:

$$\alpha_s = 1720 I^{2.28}, \quad (15)$$

where $\alpha_s$ is the dimensionless attenuation coefficient for the spherical model presented in Eq. (5) and $I$ is the local riverbed slope measured on 100 m upstream and downstream the section where the active test has been conducted. The relation above is obtained from a dataset on 14 different rivers with slope varying between 0.02-2.5%. the correlation coefficient ($R^2$) of this relation is 0.87 which shows that the local riverbed slope is a good proxy for characterizing the propagation environment in the river. The data supporting Eq. (15) are presented in the supplementary material (S2).

Figure 8 shows the global calibration curve using the cross-sectional average measured acoustic power and the inversed calibration curve using the corresponding cross-sectional average inversed acoustic power . Comparing both calibration curves shows that, when using inversed acoustic power, there is a minor decrease in variability (an increase of $R^2$ from 0.72 to 0.74), and a change of the fitted function with a lower power coefficient (decrease from 0.72 to 0.67). However, the main differences between these two calibration curves on bedload flux estimation can't be concluded from the change of the correlation coefficient $R^2$. It should be noted that using the different global calibration curves will lead to different bedload flux estimation for the same experiment (Figure 8). The main difference between these two calibrations will require investigations with field measurements.

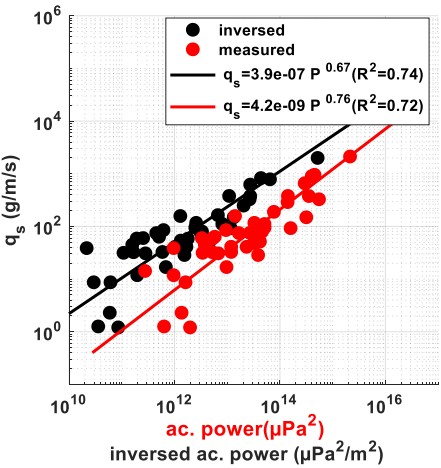

**Figure 8** Comparison of global calibration curve fitted with cross-sectional average acoustic power (in red; Nasr et al., 2023), and the readjusted curve fitted with cross-sectional average inversed acoustic power (in black)

### 6.3 Grain size detection

In lab experiments, the frequency content of the SGN has been well correlated with particle diameter (Thorne, 1985, 1986). However, in rivers, the attenuation of SGN signal at high frequency is responsible for the underestimation of bedload GSD using acoustic measurements (Geay et al., 2018). The results in Figure 8 show a noticeable correction of the inversed PSD at

high frequencies. To quantify the effect of inversion on bedload GSD estimation, the equivalent diameter $D_{eq}$ is computed by the regress empirical formula of (Thorne, 1985) as a function of central frequency:

$$D_{eq} = \frac{430}{f^{1.136}}, \quad (16)$$

Table 1 shows the computed $D_{eq}$ compared to the measured $D_{50}$ values which show that the estimated diameters using SGN measurements overestimated the measured bedload diameter. This overestimation is reduced when using the inversed source PSD. No definitive conclusion can be made on the effect of inversion model on GSD estimation using Eq. (16) as this experimental law has been carried out in controlled conditions using uniform grain-size mixtures. However, the results in Table 1 suggest a real improvement with the inversed signal. Additional effort can be made to GSD estimation by testing the model proposed by (Petrut et al., 2018) for a bedload mixture using inversed signal; however, it is beyond the scope of this article.

**Table 1 Comparison of measured bedload $D_{50}$ with estimated bedload equivalent diameter using measured and inversed PSD.**

| Experiment | 2018 | 2021 |
|---|---|---|
| Sampled $D_{50}$ (mm) | 6.8 | 3 |
| $D_{eq}$ from measured PSD (mm) | 26.8 | 12.2 |
| $D_{eq}$ from inversed PSD (mm) | 11.6 | 5.8 |

## 7. Conclusion

In this article, we present a new approach for the treatment of hydrophone measurements for bedload flux monitoring in rivers. This approach considers an inversion model for the measured acoustic profile of bedload self-generated noise SGN. The model seeks to locate the sources of SGN and calculates their power spectral density using a system of linear algebraic equations which combines acoustic measurements with acoustic signal transmission loss functions describing the propagation environment of the river.

Numerical testing shows good performance of the model with variable degrees depending on the number of separated bedload active channels in the river cross-section and uncertainty in the measured acoustic profile. Field testing of the model on the Giffre river during two very different hydraulic conditions shows that the inversion model successively corrected the attenuation of the signal PSD. The signal correction by inversion compensates for loss of acoustic power due to the propagation mainly at high frequencies. Direct bedload measurements better correlate with inversed acoustic power profiles than measured acoustic power.

The methodology presented in this paper offers new perspectives for continuous bedload monitoring with hydrophones fixed on the riverbank. Because they measure SGN for both near field and far-field, they are directly impacted by propagation effects,

and consequently, calibration is required. This calibration is possible with a reliable $qs(P)$ function associated with the drift measurement and acoustic inversion protocol.

## Appendix A: Notations:

| | | |
|---|---|---|
| $\alpha$ | frequency-dependent attenuation coefficient | $m^{-1}$ |
| $a_{m,n}$ | attenuation factor | - |
| $\alpha_\lambda$ | dimensionless attenuation coefficient | - |
| $\alpha_{\lambda c}$ | dimensionless attenuation coefficient for the cylindrical model | - |
| $\alpha_{\lambda s}$ | dimensionless attenuation coefficient for the spherical model | - |
| $\mathcal{A}$ | attenuation matrix | - |
| $\mathcal{A}^+$ | pseudo-inverse of the matrix $\mathcal{A}$ | - |
| $\mathcal{A}^-$ | inverse of the matrix $\mathcal{A}$ | - |
| $\mathcal{A}^t$ | transpose of the matrix $\mathcal{A}$ | - |
| $c_f$ | celerity of sound in water | $m \cdot s^{-1}$ |
| $D$ | particle diameter | $m$ |
| $d$ | water depth | $m$ |
| $D_{eq}$ | bedload equivalent diameter | $m$ |
| $D_{50}$ | bedload median diameter | $m$ |
| $e$ | energy spectrum density | $\mu Pa^2 \cdot s \cdot Hz^{-1}$ |
| $\epsilon$ | model error vector | - |
| $f$ | frequency | $Hz$ |
| $fc$ | central frequency | $Hz$ |
| $f_{max}$ | maximum integration frequency | $Hz$ |
| $f_{min}$ | minimum integration frequency | $Hz$ |
| $I$ | riverbed local slope | $Hz$ |
| $i$ | bedload sample index | - |
| $k$ | diameter class index | - |
| $q_s$ | specific bedload flux | $g \cdot s^{-1} \cdot m^{-1}$ |
| $\bar{q}_s$ | average specific bedload flux | $g \cdot s^{-1} \cdot m^{-1}$ |
| $M$ | number of sources | - |
| $m$ | measurement index | - |
| $N$ | number of measurements | - |
| $n$ | measurement index | - |

| $P_{\wp}$ | integrated measured power | $Pa^2$ |
|---|---|---|
| $P_{\hat{s}}$ | integrated source power | $Pa^2 \cdot m^{-2}$ |
| $P_{\widehat{s_m}}$ | integrated inversed power | $Pa^2 \cdot m^{-2}$ |
| $\overline{P_{\hat{s}}}$ | average of all sources power in the river | $Pa^2 \cdot m^{-2}$ |
| $\wp$ | measured PSD | $\mu Pa^2 \cdot Hz^{-1}$ |
| $\boldsymbol{\mathcal{P}}$ | measured PSD vector | $Pa^2 \cdot Hz^{-1}$ |
| $\boldsymbol{\widehat{\mathcal{P}}}$ | modelled PSD vector | $Pa^2 \cdot Hz^{-1}$ |
| $r$ | distance source-hydrophone | $m$ |
| s | PSD for a point source | $Pa^2 \cdot Hz^{-1}$ |
| $\hat{s}$ | source PSD per unit area | $Pa^2 \cdot Hz^{-1} \cdot m^{-2}$ |
| $\widehat{s}$ | inversed PSD per unit area | $Pa^2 \cdot Hz^{-1} \cdot m^{-2}$ |
| $\boldsymbol{S}$ | sources PSD vector | $Pa^2 \cdot Hz^{-1} \cdot m^{-2}$ |
| $\boldsymbol{\widehat{S}}$ | inversed PSD vector | $Pa^2 \cdot Hz^{-1} \cdot m^{-2}$ |
| $\sigma$ | condition number | - |
| $TL$ | transmission loss function | - |
| $TL_1$ | geometrical spreading function | - |
| $TL_2$ | scattering and absorption function | - |
| $VEcv$ | variance-explained accuracy measure | - |
| $W$ | width of the river | $m$ |
| $W_M$ | width of the sources | $m$ |
| $x_{hyd}$ | hydrophone x coordinate | $m$ |
| $y_{hyd}$ | hydrophone y coordinate | $m$ |
| $z_{hyd}$ | hydrophone x coordinate | $m$ |

## Data availability

Data supporting this article are uploaded to (Nasr, Mohamad; Johannot, Adele; Geay, Thomas; Zanker, Sebastien; Recking, Alain; Le Guern, Jules (2022), "Optimization of passive acoustic bedload monitoring in rivers by signal inversion", Mendeley Data, V1, doi: 10.17632/vygy6tsy5n.1). A supplementary material document support the findings in section 6.2 is uploaded separately on the journal website.

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
