# Peer review of "Optimization of passive acoustic bedload monitoring in rivers by signal inversion"

_Earth Surface Dynamics, 2022_

## Referee Comment (RC1)

**Overarching suggestion**

The inversion model gives a relatively simple physical-based foundation for evaluating the relationship between the acoustic source in the form of sediment impacts. To better be able to assess the model context and performance relative to other existing ones, I think that an introduction of the various models proposed thus far for the relationship between acoustic power and bedload flux should is needed (e.g., Thorne and Foden, 1988; Nasr et al., 2022). These models significantly differ from your inversion model in terms of the model derivation approaches and parameter space. E.g., the Nasr et al., (2022) model is derived using process-based bedload kinematic arguments, while the inversion model does not require such an in-detail perspective. I think you should make these points clear in your manuscript. Following your results and interpretations, you could also add a discussion about the different aspects in which the new inversion model is different than previously proposed models.

**Line-by-line comments**

Abstract: I think you should explain more about the model - what is unique about it? How is it different than the model proposed by Nasr et al., (2022)?

Line 14: 'measurements': are these measurements? Or data?

16: this sounds like you are supplying a full solution, are you?

19-21: grammar issues: (1) 'we tested the model using two field'... the field campaigns are not used to test the model, but the data are. (2) 'which measured the'... the field campaigns did not measure - the sensors measure...

22: define what the bedload profile is.

22-24: This is a little bit uninformative and ambiguous: it is unclear what a 'unique' curve is.

24: 'The inversion model shows..' - How does it show that? Perhaps the comparison of the model to data is essential for this interpretation?

27: 'Bedload…' do you mean bedload transport?

28: (1) the 'but' implies that you are contrasting the following sentence relative to the preceding one, are you? I don't think so. (2) I think that this sentence is constructed wrongly since it implies that in addition to some reasoning earlier, bedload is ALSO a consequence of morphology. However, earlier you did not supply such reasoning. Suggesting rephrasing the sentence.

29: 'velocity' - and also due to spatially heterogeneous micro-topography at the bed.

32: unclear. What is bed response?

35: high-resolution: spatial and\or temporal?

37: 'these techniques' - not all 'sensors' allow this, so you might want to relocate this sentence after the sentence introducing sensors that measure the energy that is emitted by signals...

41: grammar issues. Change 'consists in measuring directly' to 'another approach directly measures...'

48-49: This by itself can also be interpreted as increasing bedload flux (or changing turbulent structure\intensity) as a function of slope. Rephrase the sentence to make it clear.

52: with 'issue' you are implying a specific problem. If you meant it like this, please specify the problem. If you are referring to the general problem of formalizing an inversion model then state it explicitly.

53: for five gravel-bed rivers. (add 'rivers')

54-55: This sentence is misleading - the overestimation of data can suggest more views, not only that the finest grain sizes are not captured by the model.

60-63: Too lengthy and confusing. Split into two sentences.

65: 'this' - Do you mean the Geay et al., (2019) work? it is unclear.

73: you need to make it clear that the pressure fluctuations are occurring in a fluid media.

78-80: It might be a nuance, but I think it would be a result of a general physical model, isn't it? e.g., the Tsai et al., (2012) model for seismic noise contains the same properties.

98: what do you mean by source profile? Explain.

123: Define the variable h.

136: I am unclear if the attenuation coefficient is alpha_lambda as defined in line 132, at which you referred to it as a dimensionless attenuation constant'. Clarify and use the same terms.

137: 'in the first place' - This part of the sentence is unclear. Do you mean 'to first order'?

143: 'with TL calibrated with' - Grammar. Rephrase.

146: what is global in that sense? Clarify.

149: delete 'Let's'.

149: 'in the space' - why not be more specific and state that the sources are located on the river bed? if you mean that particle-to-particle impacts are also considered then you need to address these explicitly.

152: 'surfacic acoustic sources' - What are these? Define.

160: Add an opening sentence giving a rationale for the paragraph. Currently, it is unclear where you are leading it without reading it through.

165: the notation of the Figure's colors belongs to the figure's caption. Not in the main text.

169: 'the vertical green lines…' - this also belongs to the fig's captions.

179: section 3.1: This section can be better communicated. The rationale is clear, but there are some structure and communication issues that make the following paragraphs hard to follow. Start with introducing the problem and Fig. 2 as you do. Describe the settings and why the example using a drift boat is relevant to acoustic measurements. Try to better communicate the variables, their meanings, and their relevance in the specific equations.

187: change 'For solving' to 'to solve'.

187: 'the acoustic measurements' - measurements are not a parameter. Do you mean data PSD? Clarify.

188: replace 'the' with 'a' before 'drift'.

190: the x direction is not specified in your text.

190: after 'PSD', add either 'data' or 'measurements'.

204: In Eq. (9b), explain the annotations of the attenuation variables. E.g., what does $A_{1,N}$ stand for?

Also - explain the rationale of Eq. 9b. To form an intuition of this model, give a short example of the algebraic calculation followed by specific outcomes of the matrix multiplication.

212-213: 'best fit of the measured...' that does not read well - rephrase.

223: add 'a' before 'minimization'

223: add a comma after 10b

227: 'instability' - how do you define stability in that case? does the solution diverges? explain.
227: what are slight variations? Explain.

234: 'close row values' – unclear, explain.

235: which coefficients?

236: N = M – is there any method\wat to constrain the number of sources?

243-244: The sentence about bedload active channel distributions is unclear. Rephrase.

246: 'blue curve' – move to the figure's caption.

248: 'above each….' this is unclear given that you are dealing with a 3D domain. State explicitly that the above is in the vertical dimension.

248: 'simulated PSD' - You need to differentiate between 'simulated' and 'measured' - in Eq. (7) the variable P is the measured acoustic power. Here it is simulated using Eq. (7). Make the differences between an actual measurement and a simulation clearer.

251-251: move figure descriptions to the caption.

251: 'random coefficients' - this is vague. You need to give the reader the possibility to reproduce your results. What you could do is, in the supplementary material, explain in a few

sentences in what way you introduced such noise. I also encourage you to consider the sources of such noises (however, it may be better to discuss this in the discussion).

263: the imposed one is not inversed - rephrase the sentence.

269: 'that is unit\scale, …' – the sentence is unclear, rephrase.

274: How do you define 'good'? I would argue that $VE_{cv} = 0$ is good. Hence, isn't it comparative?

280: Figure 3 caption - I think that the figure should be independent, to some extent, from the text. Thus, make sure that just by looking at the figure + reading its caption, the reader is able to understand the context and the results. Specifically: 1. I don't understand what the titles (VEcv) stand for. Clarify .2 .You need to better communicate the different curves. It is a relatively complicated figure with lots of details. Extend the caption to explain the different curves (red; blue; black); 3 .What is the difference between the red curves: specifically, what is the difference between SGN with\without noise?

283: 'This first experiment' – beginning with this statement makes the readers ask themselves whether they've missed anything. In other words - context is missing. Add a sentence or two between titles 4 and 4.1 explaining what you are about to do in the following Section 4.

301-302: 'at a distance of' - unclear. Explain the location such that it is better understood.

309: define the directions of x and y in space relative to the river dimensions.

314: I am unclear about what is 'drift n' – rephrase.

321: remind the readers what is the source of the coefficient

323: 'third-octave band' – why?

326-327: 'The ratio between…' - Unclear. Rephrase sentence.

329: Figure 5b plots bedload flux rather than acoustic power. Do you mean 4b?

337: I am unclear about what is 'ex' in the parentheses.

338: 'Figure 5c' – you actually mean 4c.

338-339: move the color descriptions to the figure's caption.

341: 'lab-measured spectrum' – unclear.

342-343: I don't understand what you did here and what the point is. Clarify.

343: 'Figure 5b' – it is actually 4b.

345: 'good performance' - Again - I am unclear on how you define 'good' performance. Mentioning the spherical model association kind of makes me want to compare the performance to a different model (E.g., the cylindrical model). I understand that this is probably beyond the scope of your study, thus you need to carefully consider your phrasing here.

Figure 4 captions: the dashed red line cannot be spotted in the figure.

Additionally, you mention the mean inversed spectrum, in the red line, but I am unclear on how to recognize it.

350: replace 'The experiment' with 'An experiment'.

352: 'Two SGN and bedload flux…' - What is the difference between the two? I think you want to say here that bedload produces SGN, right?

358-359: 'similarly to the active test'… I think this sentence is redundant.

361: Fig. 6a does not show what you mean. You have a problem with figure numbering(?).

370: 'For solving the inversion problem' - be more specific and clear with your aims.

373-374: How are they estimated? Explain.

382: 'left part of the river section' – in the text, add the specific location on the x-axis.

384: 'a qualitative analysis' – unclear, what do you mean by that?

385: 'heard' - Do you mean that you are listening to the files? Clarify.

389-390: explain this interpretation.

Figure 5: in panel b, delete the zero before 2021.

395: Replace the first 'The' with 'A'.

405: Given that you are using the measured profile to conduct the inversion, I don't understand the rationale behind comparing also the measured acoustic profile.

409: Change 'The Values' to 'The values'.

410: better than what?

416: Remind the reader what 'window' stand for.

417-418: move the color and point descriptions to the figure's caption.

420: RMA is not defined. Please do.

420: This is a judgment\subjective statement. Replace with stating the reasoning for using such regression method and emit the 'recommended'.

428-429: Vague. Clarify and reference specific figures.

437: add 'the' after 'in'

440: I think you should also mention the sound generated by rainfall impacting the water surface, as well as the turbulence fluctuations near the boundaries (bed and air).

445: 'qs = f(P)' - spell out the relationship using text.

446: Separate 'as' from 'predicted' (they are attached to each other).

446: The citation of Nasr et al., lacks the associated year of publication.

453: how do you reason such a reduction of the effects by the inversion model?

454: I am unclear on how you interpret the different transport conditions. You need to be more explicit in this argument - do you mean the difference between 2018 and 20121? Do you mean the different conditions along the cross-section? Or both? Clarify.

455: I am unclear on how you separate noises from various sources following the inversion. Elaborate.

456: emit 'in this context'.

459: Slope: I think this is a great contribution made by your study which should be emphasized in the main text. You could elaborate a little bit here on the process, and present the empirical equations your derived. Also, state how Geay et al., (2019) obtained\calculated channel slope so that this is reproducible by others in the future.

462-463: move 'red' and 'black' to the figure's caption.

471: emit 'in the first place'.

476: 'most important' – that's judgment statement.

477: You have extra parentheses before the citation of Geay et al. (2018). Remove.

478: 'naïve' - judgment. Remove.

480: add 'the' before 'results'.

482: replace 'out of' with 'beyond'.

499: 'ss' is misspelled. You probably mean 'as'.

---

## Author Comment (AC1)

Reviewers' Comments and Authors Response

**Optimization of passive acoustic bedload monitoring in rivers by signal inversion**

**Mohamad Nasr[1], Adele Johannot[1], Thomas Geay[2,3], Sebastien Zanker[4], Jules Le Guern[,3], Alain Recking[1]**

[1]University Grenoble Alpes, INRAE, ETNA, Grenoble, France.

[2] Office National des Forêts, service Restauration Terrain Montagne 38000 Grenoble, France.

[3] GINGER BURGEAP, R&D, 38000 Grenoble, France.

[4]EDF Hydro, DTG, 38950 Saint-Martin-le-Vinoux, France.

*Correspondence to*: Mohamad Nasr (mohamadnasr94@gmail.com)

We thank the editorial team for giving us the opportunity to submit a revised draft of the manuscript "Optimization of passive acoustic bedload monitoring in rivers by signal inversion" for publication in the Journal of Earth Surface Dynamics. We appreciate the time and effort the reviewers dedicated to provide feedback on our manuscript and are grateful for the comments and valuable improvements to our paper. We have incorporated all the suggestions made by the reviewers. The manuscript highlights those changes (Track Change in the Word file). Please see in blue below for a point-by-point response to the reviewers' comments and concerns.

The revised manuscript contains some modifications which have not been requested by the reviewers:

- All the text and figures related to the reconstructed measured profile using the inversed profile have been deleted.
- The inversed global calibration curve has been updated using the data of Nasr et al. (2023)

**Response to reviewer comments 1**

We thank the reviewer for his time to carefully read the article. First, we present all the comments below which requires additional clarification from our side. Second, at the end of the section (Response to reviewer comments 1), there is a list of all the suggestions related to changes in the text. These suggestions have been all accepted and the text has been changed exactly as proposed by the reviewer.

**Overarching suggestion**

The inversion model gives a relatively simple physical-based foundation for evaluating the relationship between the acoustic source in the form of sediment impacts. To better be able to assess the model context and performance relative to other existing ones, I think that an introduction of the various models proposed thus far for the relationship between acoustic power and bedload flux should is needed (e.g., Thorne and Foden, 1988; Nasr et al., 2022). These models significantly differ from your inversion model in terms of the model derivation approaches and parameter space. E.g., the Nasr et al., (2022) model is derived using process-based bedload kinematic arguments, while the inversion model does not require such an in-detail perspective. I think you should make these points clear in your manuscript. Following your results and interpretations, you could also add a discussion about the different aspects in which the new inversion model is different than previously proposed models.

We thank the reviewer for this comment. As the reviewer mentioned, the model of Nasr et al. (2022) use the bedload flux and the kinematic of each bedload particle to model all impacts in the river to calculate the bedload self-generated (SGN) sources. Then, Nasr et al. (2022) propagate the signal from these sources to different positions in the river. However, the latest didn't present in their work any methodology or interest to use the inverse of the model (calculate SGN sources using the measured values). In the inversion model here we benefit from the physics behind SGN presented by Nasr et al. (2022), and we focus on the effect of propagation. We then proposed a solution for calculating SGN sources by inversing the linear algebraic relationship between measured SGN and source (Equation (9)). For that, we find that there is complementarity between these two works instead of "difference".

However, we appreciate the reviewer's suggestion that it should be more clear for the reader the why in our inversion model we don't use the kinematic of bedload particle as presented in previous works (Barton, 2006; Nasr et al., 2022). For that we modified the text at the end of section 2 to clarify this point. The revised text reads as follows:

- L179-187: "The physical model of Nasr et al. (2022) calculates the acoustic source of bedload SGN as in Eq. (1) starting from the hydraulic conditions of the river and bedload characteristics (flux and GSD). The latest then modelled the distribution of the propagated SGN in the river ($\wp$) and compared it to measured values. Nasr et al. (2022) concluded that the comparison of the modelled SGN with the measured values is highly dependent on the chosen empirical formula for impact rate ($\eta$) and velocity ($U_c$) (Eq. (1)) which are parameters difficult to validate and measure in the field. In our inversion model, we use the measured SGN ($\wp$) and the transmission loss function ($TL$) to calculate the bedload SGN source ($s$) which is independent of the propagation characteristics of the river. Equation (1) shows the dependency of the source $s$ on the bedload flux, however following the results of Nasr et al. (2022) and the limitations on measuring or estimating parameters such as bedload particles impact rate and velocity, the inversion of Eq. (1) to estimate the bedload flux directly from $s$ will not be covered in this article."

**Line-by-line comments**

Abstract: I think you should explain more about the model - what is unique about it? How is it different than the model proposed by Nasr et al., (2022)?
Please check the response above.

L14: 'measurements': are these measurements? Or data?

Thank you for this suggestion, we modified the text and the revised text reads as follows:

- L14-16: "It has been shown that this dependency of the measured SGN data on the propagation environment can significantly affect the performance of monitoring bedload flux by hydrophone techniques".

16: this sounds like you are supplying a full solution, are you?

Thank you for this comment. In this article we propose the inversion approach to locate the bedload acoustic sources and calculate the corresponding acoustic power. The physics behind the inversion indicates that the SGN sources are independent of propagation (as explained in section 2). For that we find that the work presented in this article supports the sentence "we propose an inversion model to solve the problem of SGN propagation."

19-21: grammar issues: (1) 'we tested the model using two field'... the field campaigns are not used to test the model, but the data are. (2) 'which measured the'... the field campaigns did not measure - the sensors measure...

We corrected the issues, and the revised text reads as follows:

- L 19-21: "We tested the model using data from measured bedload SGN profiles (acoustic mapping with a drift boat) and bedload flux profiles (direct sampling with an Elwha sampler) acquired during two field campaigns conducted in 2018 and 2021 on the Giffre River in the French Alps."

22: define what the bedload profile is.

We modified the text following your suggestion and the revised text reads as follow:

- L21-23: "Results confirm that the bedload flux measured at different verticals on the river cross-section correlates with the inversed acoustic power than measured acoustic power."

22-24: This is a little bit uninformative and ambiguous: it is unclear what a 'unique' curve is.

We thank you for highlighting this unclarity in the text. We replaced "unique" by "common".

24: 'The inversion model shows..' - How does it show that? Perhaps the comparison of the model to data is essential for this interpretation?

We thank you for this suggestion. We modified the text, and the revised text reads as follows:

- L24-26: "The results of the inversion model, compared to measured data, show the importance of considering the propagation effect when using the hydrophone technique and offer new perspectives for the calibration of bedload flux with SGN in rivers."

27: 'Bedload…' do you mean bedload transport?

"Bedload transport". We modified the text.

28: (1) the 'but' implies that you are contrasting the following sentence relative to the preceding one, are you? I don't think so. (2) I think that this sentence is constructed wrongly since it implies that in addition to some reasoning earlier, bedload is ALSO a consequence of morphology. However, earlier you did not supply such reasoning. Suggesting rephrasing the sentence.

We thank the reviewer for highlighting this point. We modified the text and the revised text reads as follows:

- L29-31: "Meanwhile, bedload transport is a consequence of the morphology (Recking et al., 2016) as it occurs at different rates across the channel (Gomez, 1991) due to heterogeneity in riverbed grains size distribution (GSD), riverbed geometry, flow depth, and velocity (Ferguson et al., 2003; Whiting & Dietrich, 1990)."

29: 'velocity' - and also due to spatially heterogeneous micro-topography at the bed.

Thank you for this suggestion: We added "riverbed geometry" to the indicated sentence.

32: unclear. What is bed response?

We thank the reviewer for highlighting unclarity: We modified the text and the revised text reads as follows:

- L33: "This explains why estimating bedload transport and its impact on the riverbed is not an easy task."

35: high-resolution: spatial and\or temporal?

"spatio-temporal"

48-49: This by itself can also be interpreted as increasing bedload flux (or changing turbulent structure\intensity) as a function of slope. Rephrase the sentence to make it clear.

We thank the reviewer for highlighting unclarity. We modified the text and the revised text reads as follows:

- L49-52: "For example, in their attempt to derive a general calibration curve between bedload flux and acoustic power, (Geay et al., 2020) observed that the spectral content of SGN was highly correlated to the riverbed slope which is a parameter that significantly controls the propagation environment of the river (Geay et al., 2019). (Geay et al., 2020) then suggest the significant impact of the local propagation effect of the river on the measured SGN."

52: with 'issue' you are implying a specific problem. If you meant it like this, please specify the problem. If you are referring to the general problem of formalizing an inversion model then state it explicitly.

We modified the text and the revised text reads as follows:

- L55-56: "Besides, an inversion method that estimates the entire bedload GSD curve from the measured SGN spectrum has been proposed by (Petrut et al., 2018)."

54-55: This sentence is misleading - the overestimation of data can suggest more views, not only that the finest grain sizes are not captured by the model.

We modified the text and the revised text reads as follows:

- L56-59/ "However, GSD inversion model tested on five gravel-bed rivers has overestimated the measured values in particular for the finest materials (Geay et al., 2018). The latest suggested that the acoustic power measured in rivers may not adequately capture the SGN of finest materials contained in bedload due to signal attenuation at high frequencies.

60-63: Too lengthy and confusing. Split into two sentences.

We thank the reviewer for this suggestion. We modified the text and the revised text reads as follows:

- L64-66: "To our knowledge, no studies have dealt with bedload SGN sources inversion in rivers. Despite, its evident interest for bedload monitoring that inversion would give access to the characteristics of SGN sources which can improve our understanding of the bedload characteristics and distribution in the rivers."

65: 'this' - Do you mean the Geay et al., (2019) work? it is unclear.

We modified the text to improve it's clarity.

- L68-69: "In this paper, we use the workd of Geay et al. (2019) function for developing an inversion model that gives access to the SGN sources by correcting the attenuation of the measured SGN."

73: you need to make it clear that the pressure fluctuations are occurring in a fluid media.
For SGN the acoustic wave can propagate in the water media or diffract to other media (air or riverbed). We modified the text in order to define the acoustic noise in general and not only in river, and the revised text reads as follows:

- L75-76: "Acoustic noise corresponds to minute impulsive pressure ($\mu Pa^2$) fluctuations initiated at the source position and propagated to different positions."

78-80: It might be a nuance, but I think it would be a result of a general physical model, isn't it? e.g., the Tsai et al., (2012) model for seismic noise contains the same properties.

We agree with the reviewer that SGN is the result of physical model which the acceleration of rigid body (well explained in the work of Throne and Foden (1988)). We modify the text following your comment to be more precise about this idea. The revised text reads as follows:

- L89-91: "This model shows that the energy spectrum $e$ ($\mu Pa^2 \cdot s \cdot Hz^{-1}$) of acoustic noise generated due to acceleration of rigid body is dependent on multiple parameters such as particle size, impact velocity, sediment and water mechanical properties, and position of the recording sensor with respect to the noise source."

98: what do you mean by source profile? Explain.

We thank the reviewer for highlighting unclarity. We modified the text and the revised text reads as follows:

- L101-102: "Then, bedload SGN distribution on the riverbed can be considered as a proxy of the spatial variability of bedload flux in the river cross-section."

136: I am unclear if the attenuation coefficient is alpha_lambda as defined in line 132, at which you referred to it as a dimensionless attenuation constant'. Clarify and use the same terms.
We modified the text and the revised text reads as follows:

- L138-139: "The dimensionless attenuation coefficient can then be fitted using the measurements for both the spherical ($\alpha_{\lambda\,s}$) and cylindrical ($\alpha_{\lambda\,c}$) models."

137: 'in the first place' - This part of the sentence is unclear. Do you mean 'to first order'?

"deleted" replaced by "mainly"

143: 'with TL calibrated with' - Grammar. Rephrase.

We deleted the second "with" and replaced it by with "using".

146: what is global in that sense? Clarify.

We thank the reviewer for highlighting unclarity. We modified the text and the revised text reads as follows:

- L148-149: "This hypothesis is supported by the results of Nasr et al. (20212022), which showed a better global performance of the spherical propagation model when compared to the cylindrical one for the majority of the tested rivers."

149: 'in the space' - why not be more specific and state that the sources are located on the river bed? if you mean that particle-to-particle impacts are also considered then you need to address these explicitly.

Thank you for your suggestion. we precised that it is located " on the riverbed".

152: 'surfacic acoustic sources' - What are these? Define.

We modified the text to improve it's clarity.

- L154: "However, in the case of surfacic acoustic sources distributed on the riverbed $s$"

160: Add an opening sentence giving a rationale for the paragraph. Currently, it is unclear where you are leading it without reading it through.

We appreciate this suggestion from the reviewer. We added the suggested sentence:

- L162-164: "To illustrate the attenuation of acoustic signal due to propagation, Figure1 presents the acoustic signal for a uniform square unit area acoustic source $s$ ($\mu Pa^2 \cdot Hz^{-1} \cdot m^{-2}$) in addition to the propagated signals with spherical transmission loss function to different distances."

179: section 3.1: This section can be better communicated. The rationale is clear, but there are some structure and communication issues that make the following paragraphs hard to follow. Start with introducing the problem and Fig. 2 as you do. Describe the settings and why the example using a drift boat is relevant to acoustic measurements. Try to better communicate the variables, their meanings, and their relevance in the specific equations.

We added minor modifications for the text following your suggestion.

187: 'the acoustic measurements' - measurements are not a parameter. Do you mean data PSD? Clarify.

We thank the reviewer for highlighting unclarity. We modified the text and the revised text reads as follows:

- L200: "To solve the inversion problem, the first parameter to be considered is the PSD of acoustic measurements of the bedload SGN."

190: the x direction is not specified in your text.

We modified the text to indicate that x direction is specified in Figure 2.

190: after 'PSD', add either 'data' or 'measurements'.

We modified the text following the reviewer suggestion:

- L203-205: "$N$ acoustic measurements are thus assumed to be distributed on the river cross-section (x direction), from which we compute a PSD for each measurement."

204: In Eq. (9b), explain the annotations of the attenuation variables. E.g., what does $A_{1,N}$ stand for?

We appreciate the reviewer comments however this terms are already defined in the text in L211-214: "Where $a_{m,n}$ is the attenuation factor that affects the propagated signal of source $m$ when measured by the hydrophone at position $n$. The attenuation factor $a_{m,n}$ is calculated for a surfacic source using the frequency-dependent transmission loss function $TL$:"

Also - explain the rationale of Eq. 9b. To form an intuition of this model, give a short example of the algebraic calculation followed by specific outcomes of the matrix multiplication.

We agree that such example can help the readers to better understand the text. We modified the text and the revised text read as follows:

- L225-226: "Where $\mathcal{A}$ is the attenuation. The multiplication of the $n^{th}$ raw elements of attenuation matrix $\mathcal{A}$ with the sources vector $\mathcal{S}$ corresponds to the propagation of all sources in the river to the $n^{th}$ hydrophone position."

212-213: 'best fit of the measured...' that does not read well - rephrase.

We rephrased the sentence as follows:

- L233-234: "We seek the solution $\widehat{S}$ of the vector $S$, which allows the modeled vector $\widehat{P} = \mathcal{A} \cdot \widehat{S}$ to best fit to the measured acoustic $P$ vector."

227: 'instability' - how do you define stability in that case? does the solution diverges? explain.

We thank the reviewer for such a comment. We modified the text to clarify this point as follows:

- L246-247: "The pseudo-inverse algorithm for non-square matrixes exhibits a common drawback where the solution $\widehat{S}$ may suffer from divergence (instability) under slight variations in the value of the elements of $\mathcal{A}$ or $P$."

227: what are slight variations? Explain.

We modified as follows the text to explain this point:

- L247: "Slight variations in the value of any elements of the matrix"

234: 'close row values' – unclear, explain.

We precised this term as follows:

- L254-257: "In addition, relatively high resolution of hydrophone measurements ($N \gg M$, or close measurements) will lead to matrix $\mathcal{A}$ with the close values of attenuation factor ($a_{m,n}$) at the same row, consequently, rank deficient matrix."

235: which coefficients?

We thank the reviewer for highlighting unclarity. We modified the text and the revised text reads as follows:

- L255-257: "A classical solution for such instability problems is the non-negative least square (NNLS) method, a constrained least squares problem where the values in the solution vector $\widehat{S}$ are strictly positive values."

236: N = M – is there any method\wat to constrain the number of sources?

We thank the reviewer for this comment. As we presented section 3.3, Eq. (9) is better solved under the condition N=M. For that to control the number of sources we can control the number

of drifts. The number of drifts are dependent mainly on the measured river (section width, presence of physical obstacles…)

243-244: The sentence about bedload active channel distributions is unclear. Rephrase.

We rephrased the sentence and revised text reads as follows:

- L264-269: "The total bedload active channel width —the sections with bedload transport—equals 4 m. Within the active bedload channel, the source PSD $s_m$ is computed with Nasr et al. (2021); outside $s_m$ is zero. Three different configurations of bedload transport distribution have been tested (single, dual, and triple channels) which correspond to the number of separated bedload active channels in the river cross-section (Figure 3)."

248: 'simulated PSD' - You need to differentiate between 'simulated' and 'measured' - in Eq. (7) the variable P is the measured acoustic power. Here it is simulated using Eq. (7). Make the differences between an actual measurement and a simulation clearer.
In the context of this example the simulated PSD is equivalent to the measured PSD. Even if we modeled it using Nasr et al. (2022) model, but in solving equation 9 it will replace the measured SGN. This why we didn't differentiate the symbol. In addition, the term $\widehat{P_{p_n}}$ and all related text have been deleted from the manuscript as explained at the top of this document.

251-251: move figure descriptions to the caption.

A appreciate all the suggestions from the reviewer to move details from the text to the figure caption. However, for this figure we found that it is still useful for the reader to have this information in the text. It will be easy to follow up the figure and color code. However, we also added the figure description in the caption as suggested by the reviewer.

We prefer to keep it to follow up the easily with the text.

251: 'random coefficients' - this is vague. You need to give the reader the possibility to reproduce your results. What you could do is, in the supplementary material, explain in a few sentences in what way you introduced such noise. I also encourage you to consider the sources of such noises (however, it may be better to discuss this in the discussion).
We agree with the reviewer that additional explanations are informative. To be more precise, the noise added to the modeled signal is as a form of white noise (WN) signal convolved with

the SGN signal. The generation by WN can be done using different functions that are available in different programming languages (*wgn()* in the case of matlab).

We modified the text to detail that the nature of the "random coefficients" is a white noise as follows:

- L282-L283: "The noise was added in the form white noise signal convolved with the SGN signal."

263: the imposed one is not inversed - rephrase the sentence.

We rephrased the sentence as follows:

- L284-285: "In the presence of noise, the inversed source power $P_{\widehat{\mathcal{S}_m}}$ (dashed black lines) is distinct from the generated source power profile (in blue)."

269: 'that is unit\scale, …' – the sentence is unclear, rephrase.

"To numerically assess the results, a variance-explained accuracy measure ($VEcv$) parameter is introduced (Li, 2017). The advantage of this dimensionless accuracy measure $VEcv$ that it is independent from data mean, and variance according to its definition."

274: How do you define 'good'? I would argue that VEcv = 0 is good. Hence, isn't it comparative?

We appreciate such comments to improve the clarity of the texts. We modified the text considering your comment as follows:

- L296-299:"The $VEcv$ values show that the inversion model can have good performance even in the presence of noise ($VEcv \approx 0.9 \text{ close to } 1$). However, the $VEcv$ values relatively decrease (to 0.67 and 0.58) when the number of bedload active channels increases, suggesting a higher sensitivity of the model to field uncertainty under complex bedload distribution."

280: Figure 3 caption - I think that the figure should be independent, to some extent, from the text. Thus, make sure that just by looking at the figure + reading its caption, the reader is able to understand the context and the results. Specifically: 1. I don't understand what the titles (VEcv) stand for. Clarify .2 .You need to better communicate the different curves. It is a relatively complicated figure with lots of details. Extend the caption to explain the different

curves (red; blue; black); 3 .What is the difference between the red curves: specifically, what is the difference between SGN with\without noise?

Thank you for this comment. We modified the caption considering your suggestion.

283: 'This first experiment' – beginning with this statement makes the readers ask themselves whether they've missed anything. In other words - context is missing. Add a sentence or two between titles 4 and 4.1 explaining what you are about to do in the following Section 4.

We modified the text following your suggestion and we added the following sentence:

- L308: "In this section we will present two experiments for testing and validating the inversion model."

301-302: 'at a distance of' - unclear. Explain the location such that it is better understood.

We modified the text following your suggestion and the revised text reads as follows:

L328: "The source was positioned 3 m under the water's surface."

309: define the directions of x and y in space relative to the river dimensions.

Thank you for this comment. The directions x and y are defined in Figure 2 and Figure 4a.

314: I am unclear about what is 'drift n' – rephrase.

We thank the reviewer for highlighting unclarity. We modified the text and the revised text reads as follows:

- L343-346: "The acoustic measurements have been carried out on $N$ different position on the river cross-section. For each drift $n$ located at $x_n$, we measured the power spectrum of all signals impulsion during the drift and determined the median spectrum $PSD_n$."

321: remind the readers what is the source of the coefficient

We revised the text following the reviewer suggestion as follows:

- L350-351: "The attenuation coefficients presented in Eq. (4), $\alpha_{\lambda_s} = 10^{-4}$ have been estimated following the protocol proposed by (Geay et al., 2019) during the measurements day."

323: 'third-octave band' – why?

Thank you for this comment. Applying the inversion on acoustic signal represented on each frequency band is very time consuming. This is why we use representation such as $3^{rd}$ octave band. This representation is very common in acoustic signal processing. The $3^{rd}$ octave band reduces the resolution without significant effect on the PSD characteristics. In addition, for SGN inversion, comparing the results with and without $3^{rd}$ octave band presentations didn't show any significant differences. Following your comment we modified the text as follows: "To reduce the computational load the source spectrum $s_m(f)$ are calculated using the third-octave band of the measured spectrum."

326-327: 'The ratio between…' - Unclear. Rephrase sentence.

Thank you for highlighting this unclarity. We rephrased the sentence as follows:

- L355-357: "In this case, an area correction factor was applied to the inversed results in order to compare it with the loudspeaker source signal measured in the lake. The area correction factor was calculated as the ratio between the TL function calculated as in Eq. (6b) for the inversed source area and for the loudspeaker area."

329: Figure 5b plots bedload flux rather than acoustic power. Do you mean 4b?

It is Figure 4b as indicated by the reviewer. We corrected the mistake.

337: I am unclear about what is 'ex' in the parentheses.

"e.g.,"

341: 'lab-measured spectrum' – unclear.

We modified the text to clarify this point, and the revised text reads as follows:

- L370-371: "The results show that the inversed spectrums are comparable with the reference spectrum of the source characterized in the lake, which fits within the 5%-95% percentiles on most frequencies."

342-343: I don't understand what you did here and what the point is. Clarify.

Thank you for this comment. We wanted to find a representation to estimate the residual errors of the inversion model. This why we reconstructed the measured profile using the inversed source profile. We agree with the reviewer that this representation is not very informative in the absence of comparison with other model or results (e.g., cylindrical). This is why here any everywhere in the text we deleted this representation and any corresponding text.

345: 'good performance' - Again - I am unclear on how you define 'good' performance. Mentioning the spherical model association kind of makes me want to compare the performance to a different model (E.g., the cylindrical model). I understand that this is probably beyond the scope of your study, thus you need to carefully consider your phrasing here.

Figure 4 captions: the dashed red line cannot be spotted in the figure.

Additionally, you mention the mean inversed spectrum, in the red line, but I am unclear on how to recognize it.

As explained in the response to the comment above (342-343), all points related to this comment have been deleted.

352: 'Two SGN and bedload flux…' - What is the difference between the two? I think you want to say here that bedload produces SGN, right?
We modified the text and the revised text reads as follows:

- L378-379: "Measurements of SGN and bedload flux sampling were carried out during the melting season on 13$^{th}$ of June 2018 and 6$^{th}$ of July 2021."

358-359: 'similarly to the active test'… I think this sentence is redundant.

Thank you for this comment. The indicated sentence has been deleted.

361: Fig. 6a does not show what you mean. You have a problem with figure numbering(?).

We apologize for such mistakes. All figure numbering issues have been corrected in the manuscript.

370: 'For solving the inversion problem' - be more specific and clear with your aims.

We modified the text to improve clarity. The revised text reads as follows:

- L395-396: "The geometry of the SGN sources used is similar to Figure 2 with a length extended for each source between $y$ =-150 m and $y$ =150 m which account for infinite length assumption of the SGN sources."

373-374: How are they estimated? Explain.

We modified the text to improve clarity. The revised text reads as follows:

- L397: "Two active tests following the protocol of Geay et al. (2019) have been carried out to characterize the propagation environment in the Giffre River during the two measurement days in 2018 and 2021."

382: 'left part of the river section' – in the text, add the specific location on the x-axis.

Thank you for this comment, we added "between x=0 and x=13" following you suggestion.

384: 'a qualitative analysis' – unclear, what do you man by that?

Qualitative analysis is analyzing the acoustic signal by listing to them at different frequency bands. We modified the text to improve clarity. The revised text reads as follows:

- L409: "After analyzing and listing the recordings at different frequencies, bedload SGN can be clearly heard above 800 Hz."

385: 'heard' - Do you mean that you are listening to the files? Clarify.

Please check the response for the comment 384 above.

389-390: explain this interpretation.

We explained this point in the text as follows:

- L414-416: "In addition, the attenuation of the SGN signal is more important during 2018 measurements due to more water turbulence induced by the higher flow. The higher attenuation contributes to the decrease in the measured central frequency as explained in section 2.3."

405: Given that you are using the measured profile to conduct the inversion, I don't understand the rationale behind comparing also the measured acoustic profile.   *

Thank you for this comment. The objective behind this comparison is to check which profile (measured or inversed) better explain the bedload flux profile. We can see from Figure 6 that the inversed profile is different that the measured one compared to bedload flux profile. We find it useful for the reader to visually compare all of these 3 profiles together.

410: better than what?

We rephrased the sentence in clarify the message as follows:

- L435-437: "The values of $VEcv$ are presented in Figures 6e and 6f, confirming that the inversed source profile better illustrates the bedload flux than the measured SGN profile in both experiments. However, the improvement of $VEcv$ is with less extent in the 2021 experiment than that for the 2018 experiment."

416: Remind the reader what 'window' stand for.

We modified the text following your suggestion as follows:

- L442_L445: "To study the effect of inversion on the acoustic power-bedload flux relation, the measured bedload flux value at measuring position $n$ is plotted against the corresponding value of measured acoustic power and inversed acoustic power for both 2018 and 2021 experiments (Figure 7). Depending on the experiment, we can differentiate two different trends for the measured acoustic power."

420: RMA is not defined. Please do.

We modified the text following your suggestion as follows:

- L446-448: Power laws have been fitted in Figure 7 to the measured data by applying reduced major axis regression RMA which is used when data on both axes have uncertainties (Smith, 2009). The fitted power laws presented in Figure 7 show two very distinct trends, with more than one order of magnitude of bedload flux for the same acoustic power values.

420: This is a judgment\subjective statement. Replace with stating the reasoning for using such regression method and emit the 'recommended'.
Please check the response for the comment above (420)

428-429: Vague. Clarify and reference specific figures.

We thank the reviewer for highlighting unclarity. We modified the text and the revised text reads as follows:

- L455-458: "The numerical testing in section 3.3 (Figure 3) showed that the comparison between the simulated and the inversed source profile is impacted by the presence of acoustic noise in the signal. In addition, the Isere experiments results (Figure 4b) have

shown that extraneous noise sources appeared in different positions with different intensities due to uncertainty and perturbations in the measured acoustic profile."

440: I think you should also mention the sound generated by rainfall impacting the water surface, as well as the turbulence fluctuations near the boundaries (bed and air).
In knowing the measuring conditions in the field, we can confirm that acoustic measurements have not been carried out during the important rainfall event. This also can be confirmed from the acoustic measurements.
This explains the two different fits between bedload flux and acoustic power obtained for the

453: how do you reason such a reduction of the effects by the inversion model?

The Drac River next to Grenoble City is characterized by a well propagation environment, with concentrated bedload flux channels, and transport of large gravels and pebbles. The large gravels are a source of law frequency SGN (Thorne and Foden, 1988). As explained in Figure 1b, the acoustic signal at low frequencies is less attenuated. So, for a river such as the Drac River, we have relatively exaggerated SGN measured in the rivers compared to other rivers presented in the Global calibration curve. The data for the global calibration curve of Nasr et al. (2023) shows that rivers such as the Drac River (large river with good propagation and transport of gravels and pebbles) are not well presented in the dataset used for the global calibration curve. This is why using the global calibration curve contributes to overestimation of bedload flux on this river. The idea behind the inversed global calibration curve is that the inversed acoustic power is independent from the propagation environment which reduces the bias of the inversed global calibration curve to the characteristic of the river.

454: I am unclear on how you interpret the different transport conditions. You need to be more explicit in this argument - do you mean the difference between 2018 and 20121? Do you mean the different conditions along the cross-section? Or both? Clarify.
Thank you for this comment, we modified the text to clarify this point as follows:
- L483-484: "Figure 7 shows that reducing these effects by inversing the acoustic power gives access to a better adjustment of the data obtained under different bedload transport conditions between 2018 and 2021 experiments."

455: I am unclear on how you separate noises from various sources following the inversion.

Elaborate.

The separation of noises other than SGN like hydraulic noise is similar between measured signal and inversed signal. The study of Geay (2013) showed that 2 kHz can be used as a limit

between hydraulic noise and bedload SGN for gravel-bed rivers. This limit has been used in the calibration curve of Geay et al. (2020). The same limit is also applied for the calibration curve using the inversed data.

As it is described in the text, the inversion model will recover signals which have been lost due to propagation. As it is also explained, these signals are mainly at high frequency which is mainly due bedload SGN (Thorne and Foden, 1988) and not hydraulic and turbulence noises (Geay, 2013). In this case no additional noise separation criterion is required other than the 2kHz limit.

459: Slope: I think this is a great contribution made by your study which should be emphasized in the main text. You could elaborate a little bit here on the process, and present the empirical equations your derived. Also, state how Geay et al., (2019) obtained\calculated channel slope so that this is reproducible by others in the future.

We find that a great idea to present these empirical equation and related parameters. We modified the text as suggested by the reviewer (L488-L496).

In addition, these empirical equations are presented by the thesis manuscript of (Nasr, 2023) which we referred in the revised text.

476: 'most important' – that's judgment statement.

Thank you for this comment, we deleted the indicated words and modified the text as follows:

- L514-516: "Table 1 shows the computed $D_{eq}$ compared to the measured $D_{50}$ values which show that the estimated diameters using SGN measurements overestimated the measured bedload diameter."

478: 'naïve' - judgment. Remove.

Thank you for this comment, we deleted the 'naïve' and modified the text as follows:

- L516-517: "No definitive conclusion can be made on the effect of inversion model on GSD estimation using Eq. (16) as this experimental law has been carried out in controlled conditions using uniform grain-size mixtures."

**Response to reviewer comments 2**

General comments:

1) This paper presents a method of accounting for propagation effects in the use of SGN to quantify bedload in gravel-bed rivers. Subject to technical revisions, this paper should be published.

We thank the reviewer for his/her time to carefully read the article. We answered the "specific comments" below. The "technical comments" have all been accepted as the reviewer suggested.

2) There are many instances of mixed tenses throughout the document. For example lines 320, 321, 322 swap between present, past, present

We thank the reviewer for this comment, and we apologize for this type of mistakes. We revised the text and corrected such mistakes including the indicated example.

Specific comments:

1) In section 4.1.1 no comments are made about the data acquisition/recording system used other than referring to another paper. I would like to see at least a brief description of the hardware used and basic parameters such as sampling rate.

We appreciate this suggestion. The sensitivity of the hydrophone and the sampling rate of the acquisition system have been presented in the paragraph related to characterizing the loudspeaker in the lake as follows:

- L329-332: "The emitted signals were measured at a 1 m horizontal distance from the source with an HTI-99 hydrophone (High Tech, Inc., http://www.hightechincusa.com), with a sensitivity of -199.8 dB and characterized by a flat frequency response (∓ 3dB) between 2 Hz and 125 kHz. The hydrophone was connected to the EA-SDA14 card acquisition system (RTSYS company) recording the acoustic signal in ".wav" format with a sampling frequency of 312 kHz."

The same recording materials and setup have been used in the measurements in Isere rivers. This has been mentioned in the revised text as follows:

- L340-343: "We measured the acoustic profiles every 2 m between $x = 8$ and $x = 56$, with the same hydrophone and acquisition system presented above. The protocol was identical to Geay et al. (2020), with the hydrophone mounted on a floating river board (40 cm below the water surface), and freely drifted from the bridge (drift position between $y = 2$ m and $y = 4$ m from the bridge)."

2) In section 4.1.1 I would like to see a mention of the uncertainty in position introduced by the moving hydrophone

We totally agree with the reviewer, that this type of uncertainty exists in our measurements. Estimating or quantifying these uncertainties is not possible in our case as it requires precise measurement of the hydrophone position using a GPS. However, we can notice the effect of this uncertainty as perturbations in the measured acoustic signal in Figure 4b (in red). We find that informative for the reader to mention this type of uncertainty. We modified the text as follows:

- L354-368: However, some residual sources have been modelled mainly around the active source location and at other locations in the river as in the numerical test (section 3.3) in the presence of noise). It is suspectable that uncertainty (for example due to positioning of hydrophone and loudspeaker) contributes to such residual sources as they coincide with the perturbation in the measured acoustic profile (e.g., $x = 26$ and 35 m).

**Technical Comments:**

All of the comments below have been accepted and the text has been modified as suggested.

Line 28: Remove the word 'But' and start the sentence with 'Bedload'

Line 67: 'Second', not 'secondly'

Line 69: add an s after source 'where the inversed source*s*'

Line 81: remove the word 'noise', it's implied in the acronym SGN

Line 90: add a space between 'distance' and 'r', and add 'm' after 1

Line 108: Clarify if 'water level' refers to water depth

Line 156: $r(x,y,z)$ not $r(x,y)$

Line 200: I believe that the subscript of 'z_hyd,m' or 'z_hyd,n' and again r should be listed as a function of z as well as x and y

Line 220: Should read 'there are more unknowns than equations and ....'

Line 351: I presume that the measured section is *under* the bridge not, *on* it.

Line 361: There is a reference to figure 6a. I believe this is a reference to a figure that has since been removed.

Line 368: 'window' not 'widow'

Line 370: the negative sign at the very end of the line is very easy to miss. I recommend ensuring that the entire equation is on the same line

Line 377: Remove 'the punctual measuremnets and'

Line 377: Clarify that the figure shows acoustic power

line 379: 'river' not 'rive'

Figure 5 caption: the caption referes to a), b), a), b) instead of a), b), c), d)

Line 417: Capitalize 'figure' to be consistent with the rest of the document.

Line 417: The reference is to figure 9; I believe this should be figure 7 (also line 421)

Line 446: Add a space between 'as' and 'predicted'

Line 453: Start the sentence with 'Figure 7', just remove 'on the other hand,'

Line 471: add 'of' between 'attenuation' and 'SGN'

Line 477: again remove 'on the other hand'

Line 480: 'inversed', not 'invesred'

Line 494: 'loss', not 'lost'

Line 499 'as', not 'ss'

**References**

Barton, J. S. (2006). *Passive Acoustic Monitoring of Coarse Bedload in Mountain Streams*. *August*.

Ferguson, R. I., Parsons, D. R., Lane, S. N., & Hardy, R. J. (2003). Flow in meander bends with recirculation at the inner bank. *Water Resources Research*, *39*(11), 1322. https://doi.org/10.1029/2003WR001965

Geay, T., Michel, L., Zanker, S., & Rigby, J. R. (2019). Acoustic wave propagation in rivers: an experimental study. *Earth Surface Dynamics*, *7*(2), 537–548. https://doi.org/10.5194/esurf-7-537-2019

Geay, T., Zanker, S., Misset, C., & Recking, A. (2020). Passive Acoustic Measurement of Bedload Transport: Toward a Global Calibration Curve? *Journal of Geophysical Research: Earth Surface*, *125*(8), e2019JF005242. https://doi.org/10.1029/2019JF005242

Geay, T., Zanker, S., Petrut, T., & Recking, A. (2018). Measuring bedload grain-size distributions with passive acoustic measurements. *E3S Web of Conferences*, *40*, 04010. https://doi.org/10.1051/e3sconf/20184004010

Li, J. (2017). Assessing the accuracy of predictive models for numerical data: Not r nor r2, why not? Then what? *PLoS ONE*, *12*(8). https://doi.org/10.1371/JOURNAL.PONE.0183250

Nasr, M. (2023). *Development of an acoustic method for bedload transport measurement in rivers.* Université Grenoble Alpes.

Nasr, M., Geay, T., Zanker, S., & Recking, A. (2021). A Physical Model for Acoustic Noise Generated by Bedload Transport in Rivers. *Journal of Geophysical Research: Earth Surface*, *127*(1). https://doi.org/10.1029/2021JF006167

Nasr, M., Johannot, A., Geay, T., Zanker, S., Le Guern, Jules., & Recking, A. (2023). Passive Acoustic Monitoring of Bedload with Drifted Hydrophone. *Journal of Hydraulic Engineering*. https://doi.org/10.1061/JHEND8/HYENG-13438

Petrut, T., Geay, T., Gervaise, C., Belleudy, P., & Zanker, S. (2018). Passive acoustic measurement of bedload grain size distribution using self-generated noise. *Hydrology and Earth System Sciences*, *22*(1), 767–787. https://doi.org/10.5194/hess-22-767-2018

Smith, R. J. (2009). Use and misuse of the reduced major axis for line-fitting. *American Journal of Physical Anthropology*, *140*(3), 476–486. https://doi.org/10.1002/AJPA.21090

Whiting, P. J., & Dietrich, W. E. (1990). Boundary Shear Stress and Roughness Over Mobile Alluvial Beds. *Journal of Hydraulic Engineering*, *116*(12), 1495–1511. https://doi.org/10.1061/(ASCE)0733-9429(1990)116:12(1495)

---

## Author Response (AR2)

Dear Dr. Jens Turowski,

Thank you for your thoughtful review of our manuscript. We appreciate the valuable insights you have provided. We have carefully considered all of the comments you provided and have addressed them in this document in blue

**Comment 1:**

- I have the impression that some of the physics underlying the approach are a little unclear. Specifically, you state that you build your method on the assumption of additive PSDs (lines 88-89, eq. 7). Yet, displacements (amplitudes) are additive, energy is proportional to amplitude squared, and power is the derivative of energy wrt time. So, in general there should be a non-linear relationship. It may be that this accounted for implicitly in the equations; yet, it would be good to clarify the physical relations.
- L-88 two points: 1) amplitude is additive, not power (which scales with amplitude squared), 2) the additive effect of amplitude is true regardless of the distribution of noise.

Thank you for highlighting this point. First, we would like to acknowledge that both acoustic and seismic signals are forms of wave propagation in mediums (air or water for acoustics and the Earth for seismic), which can show several similarities however we can note also some differences to be considered. Unlike seismic waves, which involve particle motion in the Earth's crust, acoustic waves in water primarily involve variations in pressure. As a result, the concept of seismic moment, which is central to seismic source characterization, does not directly apply to underwater acoustics. Instead, we work with acoustic power, which is the rate of energy transfer through the water medium due to pressure variations (which is not a mechanical force).

Both acoustic and seismic follow the superposition and interference principles for waveform combination. The latest states that when waves from multiple sources pass through the same medium, their effects sum algebraically at each point.

In our method, 'additive' pertains to the accumulation of energy contributions across multiple bedload impacts in the frequency domain, which subsequently results in the total power of the measured signal. Regarding the assumption of additive PSDs (lines 88-89, eq. 7) in our method, we acknowledge that the non-linearity in the summation of acoustic powers may not be as strict as it is in the case of seismic waves. Allow us to clarify this aspect.

In the context of underwater acoustics and signal processing, the assumption of additive power spectral density (PSD) in our method aligns with well-established principles of coherent summation. While displacement amplitudes are additive in acoustic waves, it is important to note that the relationship between power and amplitude squared introduces some non-linearity. However, when dealing with random signals in time, such as ambient noise or acoustic emissions from various sources, the central limit theorem can be assumed. The central limit theorem states that the sum of a large number of independent and identically distributed random variables tends to follow a Gaussian distribution. In the case of acoustic powers, which are derived from the squared amplitudes of individual acoustic wave components, the summation often involves numerous contributing factors. The contributing factors in this context refer to individual sources, where each of these sources contributes its own amplitude and associated power to the overall acoustic field. As a result, the central

limit theorem justifies the approximate linearity in adding acoustic powers, especially when the number of contributing factors is large (Papoulis, 1991) which is the case of bedload SGN sources (particles).

In underwater acoustics, the linear addition of acoustic powers is widely recognized as a common approach for coherent signal processing and source localization (Etter, 2018; Jensen et al., 2011). Therefore, despite the non-linearity introduced by the power-amplitude relationship, the assumption of adding acoustic powers remains a valuable and practically applicable concept in underwater acoustics and source localization (Vorländer, 2008).

Our method builds upon these established principles of underwater acoustics, coherent signal processing, and source localization.

Following your suggestion we modified (L103-L113) the text to support our assumption of linear relationship of acoustic power and energy.

**Comment 2:**

L-73 the method described in this chapter seems quite similar to methods for locating sources in seismology, especially amplitude source location (ASL). This is not surprising; after all, it is about wave propagation.

We appreciate your suggestion and will certainly incorporate in the introduction acknowledgment of prior work in the field signal inversion (seismic and acoustic). The revised text reads as follows:

- L60-L74: "The inversion method uses propagation laws to reconstruct the strengths and location of sources from the measured signal. It is extensively studied and used in acoustical engineering applications such as detecting noise sources for jet engines using a beamforming microphone array by manipulating the phase and the amplitude of the wave form (Presezniak & Guillaume, 2010), identify acoustic emissions in machinery using the spectral analysis coupled with the time-domain of acoustic signals (Arthur et al., 2017), and analyze vibrational patterns in automotive components using finite element models to reconstruct the source and propagation path (Madoliat et al., 2017). In seismology, inversion techniques have been instrumental in locating seismic sources using the amplitude source location (ASL) method (Battaglia & Aki, 2003; Walter et al., 2017), investigating microseismic events related to hydraulic fracturing using Stochastic inversion techniques (Maxwell, 2014), and understanding the structure of Earth's interior by determining the velocity distribution of the propagated waves (Rawlinson et al., 2010). Regardless of the specific field, inversion methods inherently involve modeling the propagation of signals in different environments. However, the inversed parameters and the used algorithm can widely vary depending the studied domain and the specificity of each application .In our work, the inversion is based on the spectral content of the measured bedload SGN signals propagated withing the river water column."

We also add the reference to the mentioned papers (Battaglia & Aki, 2003; Walter et al., 2017), in the introduction to highlight the similarities and differences between our method and established techniques in seismology. This will provide readers with a better context and understanding of the inspiration and background for our approach.

**Comment 3:**

L-115 please specify units for the TL function here.

We apologize for the oversight. We precised that the TL function is dimensionless. We also provided an explanation of the TL1 function in equation 2 and 3 making it clear that the function's values are dimensionless. This will help ensure clarity and proper understanding of the equations.

**Comment 4**

L-120 This is equivalent to generic descriptions of wave attenuation, used, for example, widely in seismology. Maybe this should be acknowledged.

We support the editor suggestion that this point should be acknowledged. We modified the manuscript to acknowledge the similarities in propagation laws between acoustic and seismic waves. The revised text reads as follows:

- L171-L175: The accuracy of acoustic inversion is highly contingent on the precise description of the environment and its corresponding propagation model. In oceanic acoustics, these propagation models have been rigorously investigated and are well-understood (Roh et al., 2008), allowing for precise prediction and control of acoustic signals . Remarkably, the principles of these propagation models bear notable similarity to the seismic wave attenuation phenomena used in seismology (Müller et al., 2010; Soham & Abhishek, 2016), further demonstrating their validity and utility across different disciplines.

**Comment 5**

please give dimensions without referring to a particular unit system (i.e., speed as L/T).

Thank for you for this valuable suggestion to use dimensions without specific unit systems. While we can understand the value of such suggestion in a scientific article, we will kindly not accept this suggestion. After trying your suggestions in using dimensions in section 2 and section 3 (such as $p^2.F^{-1}.A^{-1}$ for PSD) we find that this can be confusing for the reader to follow up with different variables, equations and dismissions. Mainly that many symbols have been already used in this article. On the other hand by including specific unit systems(such as $\mu Pa^2.Hz^{-1}.m^{-2}$), it helps the readers to easily follow up and comprehend the various equations and symbols.

However, we highlighted and justified our choice to the pressure unit in this article:

- L86-L87: "In underwater acoustics, the pressure is typically measured in micro-pascals (μPa), which is the standard metric unit for this field, and will be the unit of choice used within this work."

**Comment 6**

L154 unclear, what does 'surfacic' mean? Please define.

The use of the term "surfacic" is intended to emphasize sources that are distributed the surface the spatial distribution of such sources on a surface, in contrast to sources that might be located within the medium.

We added the following sentence in section 2.1 to clarify what surfacic source correspond to:

- L115-L119: "The riverbed then acts as a surfacic acoustic source which emphasizes the spatial distribution of bedload SGN noise at the surface of the riverbed.  In this case, the source power spectral density (PSD, the variation of power with frequency) per unit area $s$ (in $\mu Pa^2 \cdot Hz^{-1} \cdot m^{-2}$) is computed using a linear system that weights…"

**Other comments**

- 68 work

- 86 noise

- 117 Based on experimental work, Geay et al. proposed…

- 134 where

Thank you for highlighting all of these points. We corrected the text.

**References**

Arthur, F., Christophe, P., Thibaut, L. E. M., Quentin, L., Ea, L. V. A., & Antonio, P. (2017). Microphone array techniques based on matrix inversion. In *VKI Lecture Series STO-AVT- 287* (Issue July, pp. 1–24). Lecture Series von Karman institute for Fluid Dynamics 2017.

Battaglia, J., & Aki, K. (2003). Location of seismic events and eruptive fissures on the Piton de la Fournaise volcano using seismic amplitudes. *Journal of Geophysical Research*, *108*(B8), 2364. https://doi.org/10.1029/2002JB002193

Etter, P. C. (2018). Underwater acoustic modeling and simulation, fifth edition. In *Underwater Acoustic Modeling and Simulation, Fifth Edition*. CRC Press. https://doi.org/10.1201/9781315166346

Jensen, F. B., Kuperman, W. A., Porter, M. B., & Schmidt, H. (2011). Fundamentals of Ocean Acoustics. In *Computational Ocean Acoustics*. Springer New York. https://doi.org/10.1007/978-1-4419-8678-8_1

Madoliat, R., Nouri, N. M., & Rahrovi, A. (2017). Developing general acoustic model for noise sources and parameters estimation. *AIP Advances*, *7*(2), 025014. https://doi.org/10.1063/1.4977185

Maxwell, S. (2014). *Microseismic Imaging of Hydraulic Fracturing*. Society of Exploration Geophysicists. https://doi.org/10.1190/1.9781560803164

Müller, T. M., Gurevich, B., & Lebedev, M. (2010). Seismic wave attenuation and dispersion resulting from wave-induced flow in porous rocks — A review. *Https://Doi.Org/10.1190/1.3463417*, *75*(5). https://doi.org/10.1190/1.3463417

Papoulis, A. (1991). *Probability, Random Variables and Stochastic Processes* (McGraw-Hill Companies, Ed.; 3rd Edition).

Prezeszniak, F., & Guillaume, P. (2010). Aeroacoustic source identification using a weighted pseudo inverse method. *16th AIAA/CEAS Aeroacoustics Conference (31st AIAA Aeroacoustics Conference)*. https://doi.org/10.2514/6.2010-3725

Rawlinson, N., Pozgay, S., & Fishwick, S. (2010). Seismic tomography: A window into deep Earth. *Physics of the Earth and Planetary Interiors*, *178*(3–4), 101–135. https://doi.org/10.1016/j.pepi.2009.10.002

Roh, H.-S., Sutin, A., & Bunin, B. (2008). Determination of acoustic attenuation in the Hudson River Estuary by means of ship noise observations. *The Journal of the Acoustical Society of America*, *123*(6), EL139–EL143. https://doi.org/10.1121/1.2908404

Soham, B., & Abhishek, K. (2016). *Determination of seismic wave attenuation: A Review*. *9*(6).

Vorländer, M. (2008). *Fundamentals of Acoustics, Modelling, Simulation, Algorithms and Acoustic Virtual Reality*. Springer Berlin Heidelberg. https://doi.org/10.1007/978-3-540-48830-9

Walter, F., Burtin, A., McArdell, B. W., Hovius, N., Weder, B., & Turowski, J. M. (2017). Testing seismic amplitude source location for fast debris-flow detection at Illgraben, Switzerland. *Natural Hazards and Earth System Sciences*, *17*(6), 939–955. https://doi.org/10.5194/nhess-17-939-2017